# Exchange of water for sterol underlies sterol egress from a StARkin domain

**George Khelashvili[1,2], Neha Chauhan[3†], Kalpana Pandey[3†], David Eliezer[3], Anant K Menon[3]\***

[1]Department of Physiology and Biophysics, Weill Cornell Medical College, New York, United States; [2]Institute for Computational Biomedicine, Weill Cornell Medical College, New York, United States; [3]Department of Biochemistry, Weill Cornell Medical College, New York, United States

**Abstract** Previously we identified Lam/GramD1 proteins, a family of endoplasmic reticulum membrane proteins with sterol-binding StARkin domains that are implicated in intracellular sterol homeostasis. Here, we show how these proteins exchange sterol molecules with membranes. An aperture at one end of the StARkin domain enables sterol to enter/exit the binding pocket. Strikingly, the wall of the pocket is longitudinally fractured, exposing bound sterol to solvent. Large-scale atomistic molecular dynamics simulations reveal that sterol egress involves widening of the fracture, penetration of water into the cavity, and consequent destabilization of the bound sterol. The simulations identify polar residues along the fracture that are important for sterol release. Their replacement with alanine affects the ability of the StARkin domain to bind sterol, catalyze inter-vesicular sterol exchange and alleviate the nystatin-sensitivity of *lam2Δ* yeast cells. These data suggest an unprecedented, water-controlled mechanism of sterol discharge from a StARkin domain.

**\*For correspondence:**
akm2003@med.cornell.edu

†These authors contributed equally to this work

**Competing interests:** The authors declare that no competing interests exist.

## Introduction

Cholesterol, the 'central lipid of mammalian cells' (*Maxfield and van Meer, 2010*), is the most abundant molecular component of the mammalian plasma membrane (PM), where it represents one out of every 2–3 lipids (*Maxfield and van Meer, 2010*; *Menon, 2018*). Like many membrane lipids, it is synthesized in the endoplasmic reticulum (ER) and transported to the PM by non-vesicular mechanisms that make use of lipid transport proteins (*Holthuis and Menon, 2014*; *Wong et al., 2019*). These proteins operate as intracellular molecular ferries, achieving lipid exchange between membranes by reversibly extracting a lipid from the cytoplasmic leaflet of one membrane bilayer, encapsulating it within a binding pocket for transfer through the cytoplasm, and depositing it in the cytoplasmic leaflet of another membrane. Proteins that contain steroidogenic acute regulatory protein related lipid transfer (StART) domains constitute a major family of intracellular lipid transport proteins — the StARkin superfamily. Proteins of this family are generally soluble and able to diffuse freely through the cytoplasm, and they are implicated in moving glycerophospholipids, ceramide and sterol between cellular membranes (*Wong and Levine, 2016*; *Alpy and Tomasetto, 2005*). A new family of ER membrane proteins that have cytoplasmically disposed StARkin domains was recently identified, including six members (Lam1–Lam6) in the budding yeast *Saccharomyces cerevisiae* and three members (GramD1a–GramD1c, also termed Aster-A–C) in mammals (*Gatta et al., 2015*; *Elbaz-Alon et al., 2015*; *Murley et al., 2015*; *Sullivan et al., 2009*; *Sandhu et al., 2018*). Members of this new sub-family have one or two StARkin domains that bind sterols and catalyze sterol exchange between populations of vesicles in vitro (*Gatta et al., 2015*; *Murley et al., 2015*; *Sandhu et al., 2018*; *Jentsch et al., 2018*; *Horenkamp et al., 2018*; *Tong et al., 2018*). Lam1–Lam4 localize to ER–PM contact sites in yeast (*Gatta et al., 2015*; *Quon et al., 2018*) where they

play a role in sterol homeostasis. Thus, yeast cells that lack one or more of these proteins are hyper-sensitive to the sterol-binding polyene antibiotics amphotericin and nystatin, implying that they have alterations in PM sterol content and/or organization (*Gatta et al., 2015*; *Roelants et al., 2018*). Furthermore, they esterify exogenously supplied sterols up to 3-fold more slowly than wild-type cells, indicating a delay in some aspect of PM–ER sterol transport (*Gatta et al., 2015*; *Roelants et al., 2018*). A sterol homeostatic role has also been suggested for the mouse GramD1b (Aster-B) protein, which is highly expressed in steroidogenic organs. Thus, adrenal glands from a GramD1b knockout mouse are devoid of lipid droplets and show a severe reduction in cholesteryl ester content (*Sandhu et al., 2018*).

We recently reported crystal structures of the second StARkin domain of Lam4 (here termed Lam4S2) in *apo-* and sterol-bound states (*Jentsch et al., 2018*). The protein has an overall α/β helix-grip fold that forms a capacious binding pocket into which the sterol appears to be admitted head-first, through an aperture at one end, such that its 3-β-hydroxyl head-group is stabilized by direct or water-mediated interactions with polar residues (*Figure 1A*; *Figure 1—figure supplement 1A*). The interior surface of the binding pocket is surprisingly polar (*Figure 1—figure supplement 1B*) considering the hydrophobicity of its sterol clients, which include cholesterol, ergosterol, dehydroergosterol (DHE) and 25-hydroxycholesterol (25HC) (Results [also see *Gatta et al., 2015*; *Jentsch et al., 2018*; *Gatta et al., 2018*]). The surface of the protein near the entrance to the pocket is decorated with lysine residues, accounting for the enhanced ability of Lam4S2 to transfer sterol between anionic vesicles compared with its ability to transfer sterol between neutral vesicles (*Jentsch et al., 2018*); the entryway itself is partially occluded by a flexible loop, termed Ω1 (*Figure 1A*), whose functional importance in the StARkin family has been well-documented through mutagenesis studies (*Horenkamp et al., 2018*; *Gatta et al., 2018*; *Iaea et al., 2015*). The structures of other Lam/GramD1 StARkin domains are similar (*Sandhu et al., 2018*; *Horenkamp et al., 2018*; *Tong et al., 2018*), broadly resembling structures of other members of the StARkin superfamily except for one striking feature. The wall of the sterol binding cavity in all Lam/GramD1 StARkin domains is fractured along part of its length, not unlike one of Lucio Fontana's slashed canvases (*Candela, 2019*), exposing the sterol backbone to bulk solvent (*Figure 1—figure supplement 2A*).

We considered whether this unusual structural feature — henceforth termed 'side-opening' — might provide a mechanism to control the stability of sterol within the binding pocket. We posited that the ability to load sterol into the pocket, or to discharge it from the pocket into the membrane, might be controlled by water permeation via the side-opening. We used a combination of large-scale atomistic molecular dynamics (MD) simulations and functional tests to explore this hypothesis. Analysis of extensive ensemble and umbrella sampling MD trajectories revealed that sterol egress from Lam4S2 is associated with widening of the side-entrance to the binding pocket, penetration of water molecules into the cavity, and consequent destabilization of the bound sterol. The simulations identified several polar residues that line the side-opening to the pocket and that appear to play a critical role in the initial steps of the release process. The functional importance of these residues was validated experimentally by showing that their replacement with alanine compromises the ability of Lam4S2 to rescue the nystatin-sensitivity of *lam2Δ* yeast cells. Furthermore, these substitutions reduce the efficiency with which the purified protein is able to extract membrane-bound sterol and catalyze sterol exchange between populations of vesicles in vitro. These data suggest an unprecedented, water-controlled mechanism of sterol acquisition and discharge from a StARkin domain.

## Results

### Lam4S2 associates with the membrane via its Ω1 loop and C-terminal helix

We used atomistic MD simulations of Lam4S2 to explore the impact of the unique side-opening in Lam/GramD1 StARkin domains on the stability of bound sterol and its ability to exit the binding pocket. We chose cholesterol-bound Lam4S2 for these analyses because we (*Gatta et al., 2015*) and others (*Gatta et al., 2018*) have previously showed that Lam4S2 binds cholesterol. The protein also binds ergosterol, dehydroergosterol (DHE) (*Gatta et al., 2015*) and 25-hydroxycholesterol

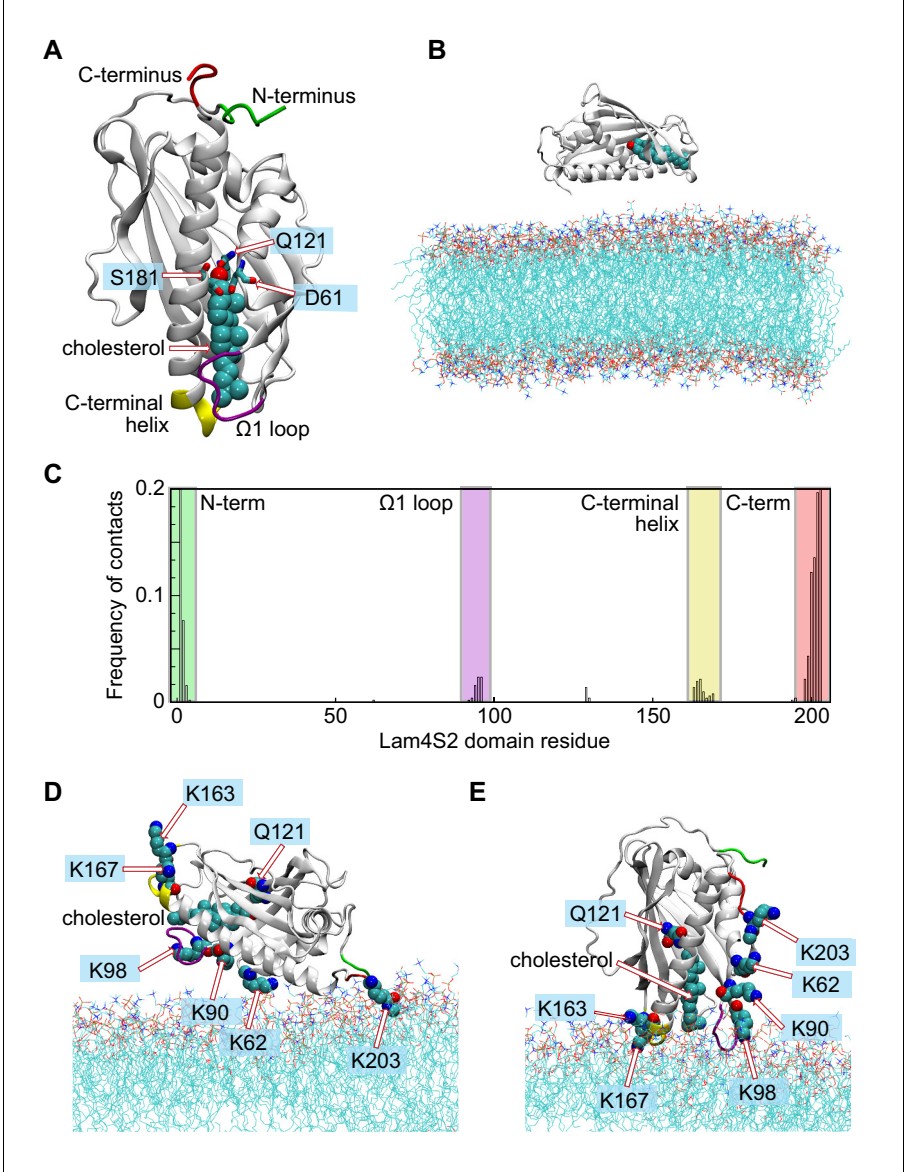

**Figure 1.** Different modes of Lam4S2-membrane association deduced from 'Stage 1' molecular dynamics simulations. (**A**) Structural elements of the Lam4S2 domain used to construct collective variables (CVs) for time-structure independent components analysis (tICA). Residues Q121, S181, D61 are labeled. Also highlighted are the locations of the Ω1 loop (purple), the C-terminal helix (yellow), the N-terminus (green), and the C-terminus (red). Cholesterol is shown in space-filling representation colored cyan except for the oxygen atom in red. (**B**) Initial positioning of Lam4S2 (cartoon) near the membrane. In this configuration, the distance between any atom of the protein and any atom of a lipid molecule was ≥10 Å. The cholesterol molecule bound to Lam4S2 is shown in space-fill representation. The water box including solution ions is omitted for clarity. (**C**) For each residue of Lam4S2, the fraction of trajectory frames from Stage 1 simulations in which the residue is in contact with the membrane was determined and plotted. A residue was considered to be in contact with the bilayer if the z-coordinate of the $C_\alpha$ atom of this residue was within 1 Å of the average z-position of the neighboring lipid phosphate atoms (identified as those located within 10 Å of this $C_\alpha$ atom). The relevant protein segments are labeled and colored using the color-code used in panel (A). (**D, E**) Two modes of Lam4S2-membrane association. The lipids in the membrane are shown as lines. The relevant protein segments are colored using the color-code used in panel (A). For completeness, panels (D, E) also show the protein-bound cholesterol molecule in space-fill, as well as residues K163, K167, K62, K90, K98, K203, and Q121.

The online version of this article includes the following source data and figure supplement(s) for figure 1:

**Source data 1.** Frequency of contacts between Lam4S2 residues and the membrane from MD simulations.

*Figure 1 continued on next page*

*Figure 1 continued*

**Figure supplement 1.** Lam4S2 binding pocket.
**Figure supplement 2.** Side opening in Lam/GramD1 structures and sterol binding specificity.
**Figure supplement 3.** Charge-mediated modes of membrane binding by Lam4S2.

(25HC) (*Jentsch et al., 2018*). All these sterols appear to bind equivalently, as the extent of DHE binding to Lam4S2 is reduced in the presence of competing amounts of cholesterol or ergosterol (*Gatta et al., 2015*), and also of 25HC (*Figure 1—figure supplement 2B*). This point is highlighted by inspection of models of Lam4S2 loaded with cholesterol (*Figure 1A*; *Figure 1—figure supplement 1B*), ergosterol (*Figure 1—figure supplement 2C*) and 25HC (*Figure 1—figure supplement 2D*) in which all three sterols are seen to be positioned in essentially the same way in the binding pocket.

Cholesterol-bound Lam4S2 (*Figure 1A*) was placed in the vicinity of a membrane bilayer (*Figure 1B*) with the phospholipid composition of the anionic 'Acceptor' liposomes used for in vitro sterol transport assays (*Jentsch et al., 2018*), and its spontaneous binding to the membrane surface was monitored via ensemble MD simulations carried out in 10 statistically independent replicates (Stage 1 ensemble simulations, 3.2 µs cumulative time). The simulations revealed two modes by which Lam4S2 associated with the membrane (*Figure 1C–E*), both of which involved the interaction of clusters of positively charged amino acids on the surface of the protein (K203 at the C-terminus, K62/K90/K98, and K163/K167 (*Figure 1D, E*)) with the overall anionic surface of the membrane. In one mode (Mode 1, occurring with ~85% frequency [*Figure 1C*]), the protein interacted with lipid headgroups via its N- and C-termini (green and red; *Figure 1D*). In the second mode (Mode 2), observed with ~15% frequency, Lam4S2 engaged with the membrane via its Ω1 loop (purple; *Figure 1C,E*) and its C-terminal helix (yellow; *Figure 1C,E*). Consistent with the overall electrostatic properties of the protein (*Figure 1—figure supplement 3A*), both modes of binding are driven by electrostatic interactions between the protein and the membrane surface. Thus, Mode 1 implicates the protein face composed of the basic C-terminus region (e.g. K203) and nearby K62 (*Figure 1—figure supplement 3A*) as well as the positively charged N-terminal amino group, whereas Mode 2 is stabilized by contacts between the membrane and basic residues K163/K167 on the C-terminal helix and K98 in the Ω1 loop region.

Although more frequent, Mode 1 is probably a non-physiological consequence of simulating the isolated Lam4S2 domain. Indeed, the N- and C-terminal regions that stabilize this mode of association are covalently linked to the remainder of the protein chain in full-length Lam4, that is, the first StARkin domain and the transmembrane helix, respectively. Furthermore, in this mode, Lam4S2 is positioned on the membrane in such a way that the hydrophobic tail of cholesterol is directly exposed to the solvent (*Figure 1D*). In Mode 2, on the other hand, Lam4S2 is engaged with the membrane via structural regions that have been implicated in the function of the protein. Thus, the Ω1 loop is known to be a functionally important feature of all StARkin domains (*Horenkamp et al., 2018*; *Gatta et al., 2018*; *Iaea et al., 2015*), and the cationic residues K163/K167 in the C-terminal helix have been implicated in the in vitro sterol transfer activities of Lam4S2 (*Jentsch et al., 2018*; *Gatta et al., 2018*) and StARD4 (*Iaea et al., 2015*). Moreover, in Mode 2, cholesterol is oriented orthogonally to the plane of membrane, with its 3-β-OH group engaging Q121 in the binding pocket of Lam4S2 and its iso-octyl tail facing the membrane (*Figure 1E*).

We hypothesized that Mode 2, in which the protein appears to be situated correctly to release sterol into the membrane, is required for proper function. To test this premise, we considered the Lam4S2 K163D/K167D double mutant, which retains an electrostatic surface that is strongly positive at the C-terminus but is negatively charged around the C-terminal helix (compare panels [A] and [B] in *Figure 1—figure supplement 3*). The double mutant binds to membrane vesicles indistinguishably from the wild type protein (*Figure 1—figure supplement 3C*), but fails to transport sterol between vesicles, as we previously reported (*Jentsch et al., 2018*). These results suggest that this mutant, like the wild-type protein, binds the membrane predominantly via the unproductive Mode 1 form of association, which is unperturbed by the mutations. However, unlike the wild-type protein, the double mutant is unable to populate Mode 2. Indeed, when the K163D/K167D mutation was introduced into the computational model of Lam4S2 representing binding Mode 2 (*Figure 1—figure*

*supplement 3D*), and the resulting system was subjected to ~50 ns MD simulations, we observed a spontaneous reorientation of the protein-membrane complex to the Mode 1 pose (*Figure 1—figure supplement 3E*). These results suggest that the K163D/K167D mutations destabilize Mode 2 and drive Lam4S2 to exclusively adopt the Mode 1 pose on the membrane surface. Importantly, given that the double mutant is functionally inactive, these observations confirm that Mode 1 is functionally irrelevant.

On the basis of these results, we chose not to explore Mode 1 further, but rather to test the premise that Mode 2 mediates release of sterol into the membrane. To this end, we enhanced the sampling of this mode of Lam4S2–membrane interaction by initiating a new set of 100 independent MD simulations (with random starting velocities) from 10 conformations of the system in which the Ω1 loop and the C-terminal helix were simultaneously engaged with lipids (Stage 2 ensemble simulations, 37.5 µs cumulative time). As described next, these trajectories revealed detailed mechanistic steps leading to spontaneous release of the protein-bound cholesterol into the membrane.

## Mechanistic steps of cholesterol transfer from Lam4S2 to the membrane

To facilitate analyses of the conformational dynamics of the membrane-bound Lam4S2–cholesterol complex in Stage 2 simulations, we used the time-structure-based independent component analysis (tICA) approach to reduce the dimensionality of the system (see 'Materials and methods'). To this end, we considered a set of collective variables (CVs) to describe the dynamics of cholesterol and relevant segments of the protein (i.e. the Ω1loop and the C-terminal helix) as well as to quantify the solvent exposure of the sterol-binding site (see 'Materials and methods' for details). All of the trajectory frames from Stage 2 simulations were projected onto the first two tICA vectors (*Figure 2A*), which represented ~90% of the total dynamics of the system (*Figure 2—figure supplement 1A*). The resulting 2D space (*Figure 2A*) was discretized for structural analyses into 100 microstates using the automated *k*-means clustering algorithm (*Figure 2—figure supplement 1B*). These microstates cover the conformational space of the system as the cholesterol molecule is transferred from the protein-bound state to the membrane.

Structural analyses of selected microstates on the tICA landscape (labeled 1–7 in *Figure 2A*), characterized by relevant CVs (*Figure 2B*) and visualized in structural snapshots (*Figure 2C*), describe key mechanistic steps of the sterol-release process. Microstate 1 represents an ensemble of states in which the sterol-binding cavity is occluded from both the solvent and the membrane. Thus, in Microstate 1 conformations (*Figure 2B,C*), cholesterol is stably bound in the protein ('chol RMSD' histogram [bottom panel, *Figure 2B*]), while the sterol-binding pocket is dehydrated ('water count' histogram [bottom panel, *Figure 2B*]) and sealed from the side by the side-chains of residues S181 and D61 that line the side-opening of the pocket ($d_{61-181}$ distance histogram [bottom panel, *Figure 2B*]). In addition, the Ω1 loop is positioned close to the C-terminal helix so that the $C_\alpha$ atoms of residues I95 in the Ω1 loop and A169 in the C-terminal helix are within ~10 Å of each other ($d_{95-169}$ distance histogram [bottom panel, *Figure 2B*]; see the middle structure in *Figure 2C* for the location of A169), thus occluding the sterol-binding pocket from below, that is the vantage point of the membrane. Indeed, in the ensemble of conformations representing Microstate 1, cholesterol has essentially no contact with membrane lipids ('number of lipids' histogram [bottom panel, *Figure 2—figure supplement 2A and B*]).

The first step in the sterol-release process involves widening of the side-entrance to the sterol-binding site, which is enabled by gradual separation of the side-chains of residues D61 and S181. This structural change on the tICA landscape can be followed in the evolution of the system from Microstate 1 to Microstates 2 and 3 (see $d_{61-181}$ distance histogram [*Figure 2B*]). Concomitant with the widening of the side-opening, the level of hydration (water count) of the binding pocket progressively increases (*Figure 2B,C*).

Cholesterol remains stably bound throughout these initial events (cholesterol RMSD is unchanged in Microstates 1–3). However, the rising level of hydration in the binding site results in destabilization of the polar interactions between the 3-β-OH group of cholesterol and the side-chain of residue Q121 as cholesterol initiates its translocation towards the membrane. Indeed, as the system transitions from Microstate 3 to Microstate 4, the RMSD of the cholesterol molecule increases (*Figure 2B*). Correspondingly, the minimum distance between the cholesterol oxygen and residue Q121 increases by ~4 Å (compare $d_{chol-121}$ for Microstates 1 and 4, *Figure 2—figure supplement*

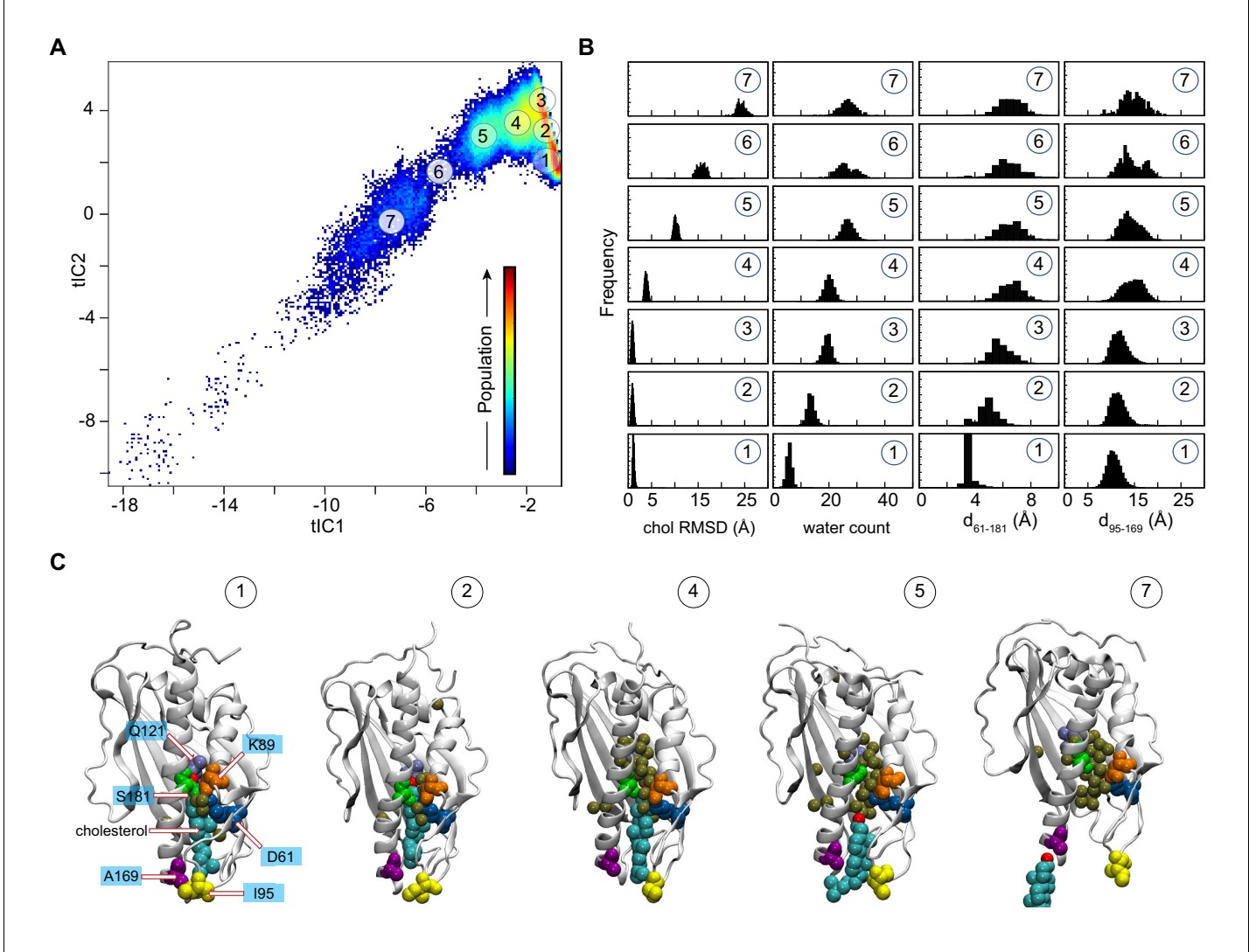

**Figure 2.** Mechanistic steps of cholesterol release from Lam4S2 revealed by tICA analysis. (**A**) 2-D landscape representing all of the Stage 2 MD trajectories mapped with the tICA transformation in the space of the first two tICA eigenvectors (tIC 1 and tIC 2). The lighter shades (from red to light green to yellow) indicate the most populated regions of the 2D space (see the color bar). Microstates representing the most populated states in these simulations are indicated by the numbered circles (1-7) and represent various stages in the lipid translocation process. (**B**) Characteristics of the selected microstates. The columns record the probability distributions of the cholesterol RMSD, number of water oxygens in the sterol-binding pocket, and distances between residues 61 and 181 ($d_{61-181}$) and between residues 95 and 169 ($d_{95-169}$). (**C**) Structural models representing selected microstates. In these snapshots, Lam4S2 is shown in cartoon, and cholesterol as well as selected protein residues (Q121, D61, K89, S181, I95, A169) are shown in space fill (the residues are labeled in the snapshot of Microstate 1). Water oxygens in the sterol-binding site are drawn as gold spheres.

The online version of this article includes the following source data and figure supplement(s) for figure 2:

**Source data 1.** Characteristics of different microstates from tICA approach.
**Figure supplement 1.** tICA analysis of Stage 2 simulations.
**Figure supplement 2.** Mechanistic steps of cholesterol release from Lam4S2 revealed from tICA analysis.
**Figure supplement 3.** Probability distribution of the number of water oxygens in the sterol-binding pocket calculated from analysis of Stage 1 ensemble MD simulations of *apo* Lam4S2 (based on PDB ID 6BYD).
**Figure supplement 4.** Sterol release sampled in multiple trajectories.
**Figure supplement 5.** Selection of wild-type trajectory frames to initiate simulations of D61A, S181A, and K89A mutants.
**Figure supplement 6.** Cholesterol destabilization during unbiased ensemble MD simulations of the S181A, K89A and D61A Lam4S2 mutants.

*2B*). Notably, as the cholesterol molecule assumes this new position, the distance between the Ω1 loop and the C-terminal helix increases, as seen in the broadening of the $d_{95-169}$ histogram (*Figure 2B*), indicating initial opening of the sterol-binding pocket towards the membrane.

Cholesterol egress then proceeds through Microstates 5–7 in which the sterol-binding pocket remains open and solvated, while the Ω1 loop continues to sample conformations that position it relatively far from the C-terminal helix (*Figure 2B,C*). The cholesterol molecule leaves the binding pocket with its tail 'down', becoming gradually encapsulated by the hydrophobic chains of neighboring lipids until it fully embeds into the lipid membrane ('number of lipids' histogram [*Figure 2—figure supplement 2B*] and corresponding structural snapshots [*Figure 2—figure supplement 2D*]). The process of translocation is complete when the system reaches Microstate 7. The remaining part of the tICA space (corresponding to lower tIC1 and tIC2 values, that is, the bottom left region of the 2D space in *Figure 2A*) describes trajectory data in which Lam4S2 disengages from the membrane, after the release of the sterol, and diffuses into the solvent. Of note, the high level of hydration of the binding site in empty Lam4S2, that is after sterol egress, is recapitulated in MD simulations of the *apo* Lam4S2 system (initiated from the sterol-free Lam4S2 structure, PDB ID 6BYD) run under the same conditions as the Stage 1 simulations of cholesterol-bound Lam4S2 described in *Figure 1* (see *Figure 2—figure supplement 3* and 'Materials and methods' for more details).

The sterol translocation process outlined above was sampled in its entirety in 5 out of 100 Stage 2 simulations (trajectories highlighted in red in *Figure 2—figure supplement 4*). Another 19 trajectories in this set sampled the evolution of the system from Microstate 1 to Microstate 5 (trajectories marked with a green star in *Figure 2—figure supplement 4*), but on the time scales of these simulations, the system either did not progress further (i.e. to Microstates 6 and 7) or returned to Microstate 4 or 3 where it remained (see also below). In the remaining Stage 2 simulations, the system fluctuated between Microstates 1, 2, and 3 (unmarked trajectories in *Figure 2—figure supplement 4*).

## The side-opening to the sterol-binding pocket is a key structural element of the release mechanism

The MD simulations indicate that widening of the side-opening to facilitate water penetration into the binding site (*Figure 3*) is a key step in the mechanism by which bound cholesterol leaves the protein to enter the membrane. To investigate in more detail the interplay between increased hydration of the sterol-binding pocket, widening of the side-entrance to the binding cavity, and stability of cholesterol within the pocket, we analyzed the dynamics of D61 and S181 and their interactions with other residues in the binding site during the simulations. We found that D61 is engaged in electrostatic interactions with residue K89 located in the β2 strand preceding the Ω1 loop. Thus, the side-chain of K89 faces the entrance of the binding pocket where it interacts with the anionic side-chain of D61 (*Figure 2—figure supplement 2C*, *Figure 3*). This interaction is maintained in the initial stages of the translocation process (Microstates 1–4), but becomes unstable as the hydration of the sterol-binding pocket reaches its highest levels after cholesterol leaves the site (*Figure 3—figure supplement 1*; also note sampling of a wide range of $d_{61-89}$ distances for Microstates 5–7 in *Figure 2—figure supplement 2B*). These data suggest that D61, S181 and K89 together participate in stabilizing the closed conformation of the side-entrance to the binding pocket.

On the basis of these results and considering the position of the K89 side-chain near the protein-solvent interface, we hypothesized that replacing the polar and relatively long side-chain of K89 with a smaller hydrophobic moiety would promote widening of the side entrance, leading to destabilization of cholesterol in the binding pocket. Likewise, substituting D61 and S181 with residues with smaller-sized hydrophobic side-chains should also have a destabilizing effect on bound cholesterol.

## Substitution of residues D61, S181, and K89 by Ala promotes hydration of the binding site and destabilizes bound cholesterol

To test these hypotheses, we computationally generated K89A, D61A, and S181A point-mutants of Lam4S2, and probed their dynamics using atomistic MD simulations. Specifically, we considered two snapshots taken at different time points (120 ns and 150 ns, respectively) from one of the Stage 2 trajectories (350 ns-long) of the wild-type protein system in which sterol release was observed. For the wild-type protein at these time points, the side-opening to the pocket is closed (*Figure 2—*

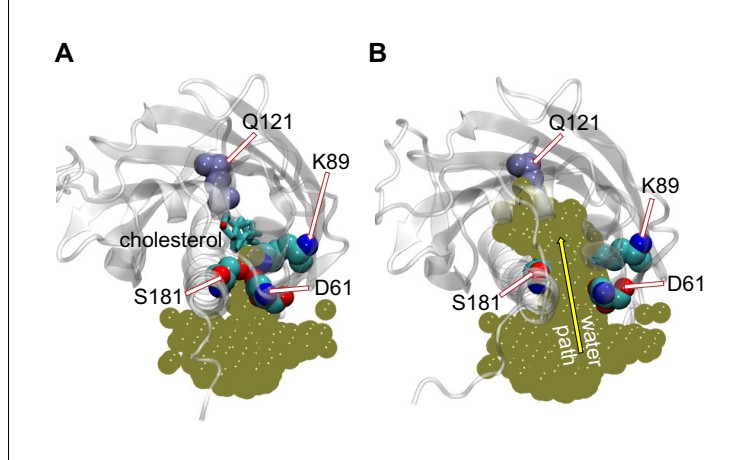

**Figure 3.** Penetration of water into the binding pocket through the side-opening is a key step in the sterol-release process. (A, B) Top view of the sterol-binding pocket in Lam4S2, illustrating closed (A) and open (B) conformations of the side-opening to the binding site (the protein models are representative structures from Microstates 1 and 7, respectively). In both snapshots, residues D61, K89, and S181 (which line the side-opening) and residue Q121 (which coordinates the cholesterol hydroxyl group) are highlighted (in space-fill and labeled). The gold spheres in panels (A) and (B) represent the superposition of water oxygens in the binding site and near the side-entrance from one of the Stage 2 trajectories before (panel A) and after (panel B) the side-entrance opens. In panel (A), the cholesterol is shown as licorice. A water pathway to the binding pocket, formed when the side-entrance is open, but not when it is closed, is illustrated in panel (B) by the yellow arrow.

The online version of this article includes the following figure supplement(s) for figure 3:

**Figure supplement 1.** Minimum distances between cholesterol and residue Q121 ($d_{chol-121}$, black), between residues D61 and K89 ($d_{61-89}$, blue), and between residues D61 and S181 ($d_{61-181}$, red) as a function of time in the five MD simulation trajectories from Stage 2 in which full release of the sterol molecule was observed.

---

figure supplement 5A), the cholesterol molecule is stably bound (*Figure 2—figure supplement 5B*), and the level of hydration is relatively low (between 5–10 water molecules as shown in *Figure 2—figure supplement 5C*). We introduced the three mutations separately into these two snapshots, and (for each construct) carried out 150-ns-long unbiased MD simulations in 10 replicates (1.5 μs total simulation time). Analysis of these trajectories revealed that for all three of the mutants, the hydration level of the sterol-binding site increased rapidly during the initial 4–5 ns of the simulations (*Figure 2—figure supplement 6A*). Note that in the original wild-type trajectory, reaching the same high level of hydration (>20 water molecules, *Figure 2—figure supplement 6C*) required a considerably longer time (~180 ns, *Figure 2—figure supplement 5C*). Furthermore, in the trajectories for the mutant proteins, cholesterol was destabilized in its binding pocket (*Figure 2—figure supplement 6B,D*). On the simulation timescales, rapid destabilization was especially notable for the K89A system in which, for all but one replicate, the sterol was unstable in its binding site (panel labeled K89A, *Figure 2—figure supplement 6D*).

## K89A-Lam4S2 has a lower energy barrier for cholesterol release

To address the effect of the K89A mutation on cholesterol stability quantitatively, we compared the energetics of sterol release in the K89A mutant versus the wild-type system using umbrella sampling MD simulations. We constructed the potential of mean force (PMF) for cholesterol release by constraining the z-distance between the sterol hydroxyl oxygen and the $C_\alpha$ atom of residue Q121, $d_{Z(chol-121)}$ to different values in the range $\in [2\text{Å}; 20\text{ Å}]$ along the release pathway ($d_{Z(chol-121)}$ histogram (*Figure 2—figure supplement 2B*)). The results are shown in *Figure 4A*. For the wild-type system, the PMF calculations indicate that cholesterol release requires that an energy barrier of ~6 kcal/mole is overcome, and proceeds through two major steps that were also identified in our tICA analysis of Stage 2 simulations. Thus, the PMF has a global minimum at $d_{Z(chol-121)}$ ~2 Å, corresponding to the position of cholesterol in the binding site where its polar head-group is coordinated by residue Q121 (snapshot at the top right of *Figure 4A*), and two local minima (LM-1 and LM-2) at $d_{Z(chol-121)}$

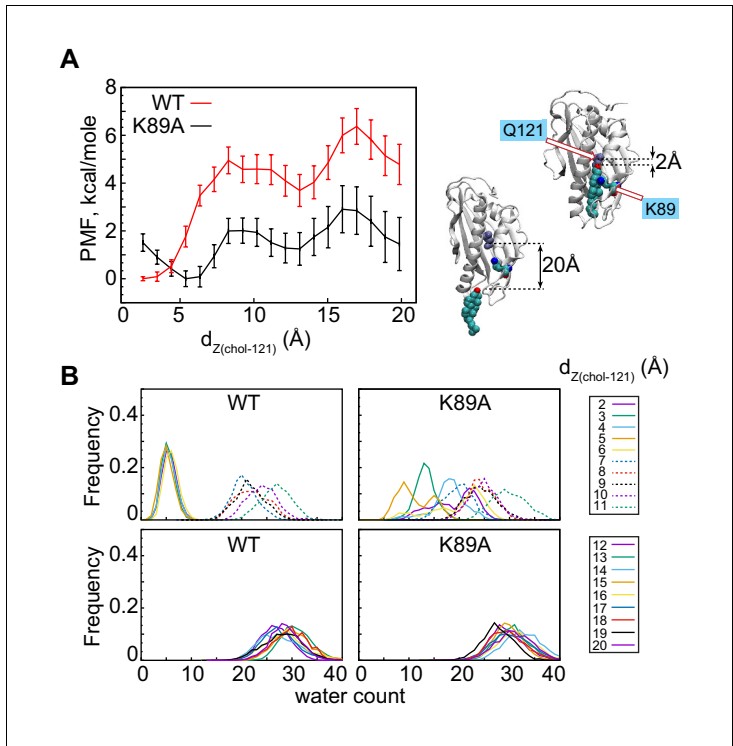

**Figure 4.** The K89A mutation reduces the energy barrier for cholesterol release. (**A**) Potential of mean force (PMF) as a function of $d_{Z(chol-121)}$ distance for wild-type (red) and K89A (black) Lam4S2, calculated from umbrella sampling MD simulations at each $d_{Z(chol-121)}$. The structural representations on the right side of the panel illustrate locations of cholesterol corresponding to $d_{Z(chol-121)}$ ~2 Å (top) and $d_{Z(chol-121)}$ ~20 Å (bottom). Residues Q121 and K89 in these snapshots are also shown. (**B**) Histograms of the number of water oxygens in the sterol-binding site constructed from analysis of trajectories representing various windows in the range of $d_{Z(chol-121)} \in$ (2Å; 20Å) from the umbrella MD simulations of the wild-type (left panels) and K89A (right panels) systems.

The online version of this article includes the following source data and figure supplement(s) for figure 4:

**Source data 1.** Potential of mean force for wild-type and K89A Lam4S2.
**Source data 2.** Water counts in wild-type and K89A Lam4S2 corresponding to different umbrella windows.
**Figure supplement 1.** The K89A mutation promotes opening of the side-opening to the binding pocket and the influx of water.
**Figure supplement 2.** Potential of mean force for sterol egress into water versus the membrane.
**Figure supplement 3.** The potential of mean force calculations with the weighted histogram analysis method (WHAM) (*Grossfield, 2013*).

~10–14 Å and > 18 Å, respectively. The global minimum represents the ensemble of states found in Microstate 1-3 ($d_{Z(chol-121)}$ histogram, *Figure 2—figure supplement 2B*), whereas LM-1 corresponds to the ensemble of states found in Microstate 5 ($d_{Z(chol-121)}$ histogram, *Figure 2—figure supplement 2B*).

The PMF calculations reveal that the energy barrier that separates LM-1 from the global minimum is ~5 kcal/mole (red trace in *Figure 4*). This high energy cost is associated with the clear change in hydration of the sterol-binding site and the concomitant opening of the side-opening to the pocket (see WT profiles in *Figure 4B*). Indeed, the water count increases and the D61–S181 interaction is destabilized when the system transitions from $d_{Z(chol-121)} \in$ [2 Å; 6 Å] to $d_{Z(chol-121)} \geq 7$ Å (*Figure 4*, *Figure 4—figure supplement 1A, C*). LM-2 represents the ensemble of states in which cholesterol is on the verge of exiting the protein, that is, states in which cholesterol is mostly engaged by lipids and with its head-group on the level of the Ω1 loop (see Microstate 6 in *Figure 2—figure supplement 2D*). LM-1 and LM-2 are separated by an energy barrier of ~2 kcal/mole. Overall, the presence of multiple minima on the PMF plot is consistent with our findings from the tICA analysis of the unbiased MD simulations described above: in some of the Stage 2 trajectories, the system evolved from

Microstate 1 to Microstate 5 (i.e. transitioned from the global minimum to LM-1 on the PMF plot), but either did not progress further to complete sterol egress (i.e. they did not reach LM2) or returned to the conformational space of the tICA landscape characterized by relatively low hydration of the sterol-binding pocket (i.e. they returned to the global energy minimum). For comparison, we also investigated the energetics of sterol release from wild-type Lam4S2 in water. In this case, the PMF profile no longer has LM-1 and LM-2 minima (*Figure 4—figure supplement 2A*). Instead, only one global minimum can be found, corresponding to the position of cholesterol in the binding site. The energy difference between the bound and released states on the PMF profile in water is >10 kcal/mole, at least ~2-fold higher than the energy barrier for sterol release into the membrane. These differences can be explained by the different solvent exposure of the hydrophobic tail of cholesterol during the egress process in water versus that in the membrane (*Figure 4—figure supplement 2B*).

Remarkably, comparison of the PMF plots for the wild-type and the K89A systems (*Figure 4A*) reveals that the mutation significantly lowers the barriers that must be overcome in order for transition between the different energy minima to occur. Thus, although the PMF profile for the K89A construct still has three energy minima, the energy cost of transitioning between the global minimum and LM-1 in this system is ~2 kcal/mole, and between LM-2 and LM-3 is ~1 kcal/mole, resulting in an energy barrier of only ~3 kcal/mole for the entire release process ($d_{Z(chol–121)} \in$ [2 Å; 20 Å]). This is approximately half of that determined for the wild-type system. This reduction in the energy cost can be explained by the greater extent of hydration of the binding site in K89A compared to that in the wild-type protein. Indeed, in wild-type Lam4S2, the binding site remains largely dehydrated until cholesterol disengages from Q121 ($d_{Z(chol–121)} \in$ [2 Å; 6 Å]) (*Figure 4B*, *Figure 4—figure supplement 1A*), whereas in the K89A protein, the level of solvation of the binding pocket is relatively high (>10 water molecules) even when cholesterol is interacting with Q121 (*Figure 4B*, *Figure 4—figure supplement 1B*). These trends in binding-site hydration are mirrored by changes in the $d_{61–181}$ distance along the $d_{Z(chol–121)}$ coordinate (note destabilization of D61–S181 interactions in the K89A system versus those in the wild-type system for small $d_{Z(chol–121)}$ values in *Figure 4—figure supplement 1C, D*).

Interestingly, the global minimum on the PMF profile of the K89A mutant is shifted compared to its location on the PMF plot of the wild-type system, from $d_{Z(chol–121)}$ ~2 Å to ~5 Å (*Figure 4A*). We found that, at the shortest $d_{Z(chol–121)}$ distances, cholesterol–Q121 interactions in the mutant are mostly mediated by water molecules, whereas at $d_{Z(chol–121)}$ ~5 Å, the hydroxyl group of cholesterol is in direct contact with Q121 (see sharp peak at ~2 Å for the $d_{Z(chol–121)}$ = 5 Å plot in *Figure 4—figure supplement 1B*; note that $d_{Z(chol–121)}$ is the Z-distance between the hydroxyl and the $C_\alpha$ of Q121). This may also explain why the water content in the cavity is skewed towards lower values for $d_{Z(chol–121)}$ ~5 Å (*Figure 4B*). Thus, for both the wild-type and K89A systems (and for the wild type simulated in water, *Figure 4—figure supplement 2A*), the global minimum on the PMF plot corresponds to the ensemble of states in which cholesterol is engaged in direct interactions with Q121. Overall, the PMF calculations reveal that the K89A substitution lowers the energy barrier for cholesterol release from Lam4S2 into the membrane and suggests that cholesterol is consequently less stable in the binding pocket.

## Alanine substitution of residues at the side-entrance to the sterol-binding pocket impacts the function of Lam4S2 in cells and in vitro

Our computational studies indicate that substitution of D61, K89 or S181 with alanine affects the degree of hydration of the sterol-binding pocket and the stability of bound sterol, with the most significant effects seen for the K89A mutant. We tested the functionality of K89A and the other mutants using three types of experiments.

We previously showed that yeast cells lacking Ysp2/Lam2 (*lam2Δ* cells) are sensitive to the polyene antibiotic amphotericin B, and that this phenotype can be corrected by expression of a soluble GFP–Lam4S2 fusion protein (*Gatta et al., 2015*). We verified that this was also the case for nystatin, another polyene antibiotic (*Figure 5A*, compare the first two rows in which *lam2Δ* cells are transformed with either an empty vector (row 1) or a vector for expression of GFP–Lam4S2 (row 2, WT), and plated on media without nystatin or with different amounts of nystatin). We then tested the ability of GFP-fused Lam4S2 proteins carrying either K89A, D61A, or S181A single-point mutations (GFP–Lam4S2(K89A), GFP–Lam4S2(D61A), GFP–Lam4S2(S181A), respectively) to rescue the nystatin

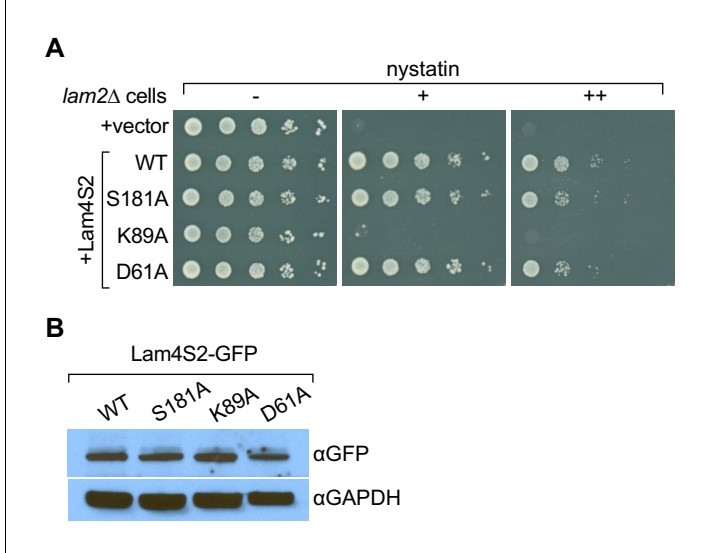

**Figure 5.** Lam4S2(K89A) does not rescue the nystatin sensitivity of *lam2Δ* cells. (**A**) Cells (*lam2Δ*) were transformed with an empty vector (top row) or with a vector for the expression of GFP–Lam4S2 wild-type (WT) or point mutants as indicated. Serial 10-fold dilutions were spotted onto agar plates containing defined minimal media (lacking [–] or containing 2 μg/ml [+] or 8 μg/ml [++] nystatin). The plates were photographed after 72 hr at room temperature. (**B**) Cell-equivalent amounts of cytosol from *lam2Δ* cells expressing GFP–Lam4S2 wild-type or mutants were analyzed by SDS-PAGE and immunoblotting using anti-GFP antibodies to detect the fusion proteins and anti-GAPDH as a loading control.

sensitivity of the *lam2Δ* cells. *Figure 5A* shows that *lam2Δ* cells expressing GFP–Lam4S2(K89A) remained nystatin-sensitive, whereas those expressing GFP–Lam4S2(D61A) or GFP–Lam4S2(S181A) became resistant to the antibiotic, similar to *lam2Δ* cells expressing wild-type protein. As all the Lam4S2 variants tested were expressed at equivalent levels (as revealed by SDS-PAGE immunoblotting [*Figure 5B*]), this cell-based assay indicates that the K89A mutant has a functional deficit, whereas the D61A and S181A proteins are able to provide cells with sufficient functionality to rescue their nystatin-sensitivity phenotype.

To test explicitly the ability of the mutants to extract sterol from membranes and to catalyze sterol exchange between populations of vesicles, we expressed His-tagged versions of the proteins in *Escherichia coli* and purified them by affinity chromatography and size exclusion. The D61A mutant proved unexpectedly problematic on account of its low yield and apparent instability, and so we focused on the S181A and K89A mutants (*Figure 6A*). Like wild-type Lam4S2, these mutants displayed monodisperse profiles on size exclusion (*Figure 6B*) and yielded circular dichroism spectra that were indicative of well-folded structures (*Figure 6C*).

Sterol extraction assays were performed by incubating the purified proteins with large, unilamellar vesicles containing [³H]cholesterol, and determining the amount of radioactivity and protein in the supernatant after ultracentrifugation to pellet the vesicles. Relative to the wild-type protein, the S181A mutant extracted only ~50% of sterol under our standard incubation conditions, whereas the K89A mutant had essentially no ability to extract sterol (*Figure 6D*).

To probe the sterol transfer activity of the Lam4S2 mutants, we performed in vitro sterol transport assays, as previously described and depicted schematically in *Figure 7A*. Donor vesicles containing fluorescent dehydroergosterol (DHE) were incubated with acceptor vesicles containing the Förster resonance energy transfer (FRET) acceptor dansyl-PE. Excitation of DHE results in sensitized fluorescence emission from dansyl-PE only when the two lipids are in the same vesicle. *Figure 7B* (see also *Figure 7D*) shows that under our standard conditions, the wild-type protein increases the rate of DHE exchange ~7 fold over the spontaneous rate. The performance of the S181A mutant was similar to that of the wild-type protein, whereas the K89A mutant had essentially no activity (*Figure 7C,D*).

Overall, the three functional tests described above indicate that the K89A mutant is compromised in sterol handling—it is unable to extract sterol from membranes or to transfer it between vesicles,

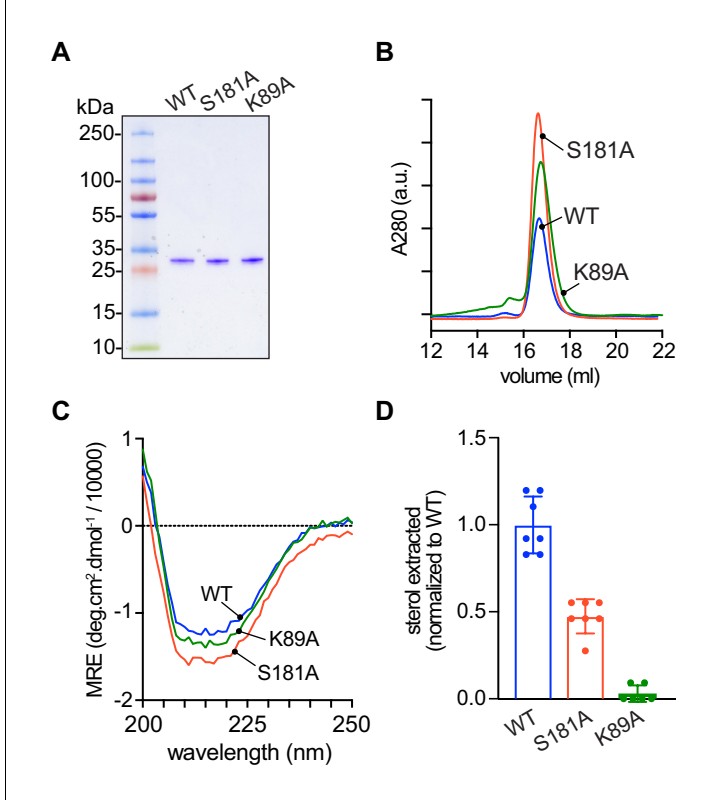

**Figure 6.** Purification and characterization of Lam4S2 mutants. (**A**) Lam4S2 wild-type and the S181A and K89A mutants were purified as His-tagged proteins via affinity chromatography and size-exclusion. The purified proteins were analyzed by SDS-PAGE (4–20% gradient gel) and Coomassie staining. (**B**) Size-exclusion analysis of purified proteins. (**C**) Circular dichroism spectra of purified proteins. Protein samples were 12 µM and the spectra shown are the average of three scans per sample. (**D**) Sterol extraction by purified Lam4S2 and mutants. Sucrose-loaded liposomes (DOPC:DOPE:DOPS:cholesterol, 49:23:23:5 mol %, doped with [3H]cholesterol) were incubated with 750 pmol of purified proteins for 1 hr at room temperature. After ultracentrifugation, the radioactivity and the protein amount in the supernatant were determined, and the stoichiometry of binding was calculated. Data are represented as mean ± SEM (error bars; n = 5–7). Data are normalized to the average value obtained for the wild-type protein (0.11 ± 0.02 pmol cholesterol/pmol protein [mean ± standard deviation (n = 6)]). The ability of the three proteins to interact with membranes was comparable (percentage bound to membranes: 42 ± 7.2 (WT), 43 ± 3.0 (S181A), 49 ± 4.5 (K89A) (mean ± standard deviation [n = 8]).

accounting for its inability to rescue the nystatin sensitivity of *lam2Δ* cells. These functional outcomes are in line with our computational prediction that cholesterol would be unstable in the binding site of the Lam4S2 K89A. Interestingly, the partial inability of the S181A mutant to extract cholesterol did not affect its ability to catalyze sterol exchange or to rescue the nystatin sensitivity of *lam2Δ* cells.

## Discussion

Lam/GramD1 StARkin domains specifically bind sterols (*Gatta et al., 2015*), admitting and exporting the sterol molecule through an aperture at the end of their long axis as suggested by inspection of crystal structures (*Sandhu et al., 2018*; *Jentsch et al., 2018*; *Horenkamp et al., 2018*; *Tong et al., 2018*) and as also seen in the MD simulations reported here. Strikingly, the sterol-binding pocket in these proteins is fractured along part of its length, exposing bound sterol to solvent. The analyses presented here describe a potentially general mechanism by which sterol egress (or entry) from Lam/GramD1 StARkin domains is controlled by the concomitant entry (or egress) of water molecules via this unusual lateral fracture.

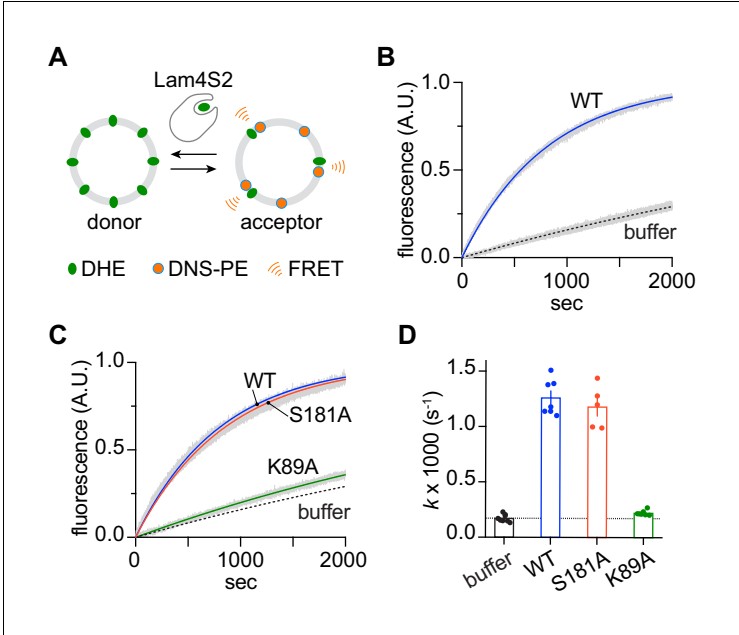

**Figure 7.** Sterol transfer activity of Lam4S2 mutants. (**A**) Schematic of the sterol transport assay. (**B**) Spontaneous sterol exchange between vesicles, and transport catalyzed by wild-type Lam4S2 (0.05 μM). Traces (n = 7–8) were acquired from three independent experiments and averaged. The blue and dashed lines represent mono-exponential fits of the averaged data; the gray bars graphed behind the fits represent the standard error of the mean (SEM). (**C**) As in panel (**B**), except that Lam4S2 mutants were tested (n = 5–7). The data fits for traces corresponding to spontaneous transport and transport catalyzed by the wild-type protein are taken from panel (**B**) and shown for comparison. (**D**) Rate constants (colored symbols) obtained from mono-exponential fits of individual traces from the experiments depicted in panels (**B**) and (**C**). The bars show the mean and SEM of the data.

Our MD simulations reveal that Lam4S2 docks onto anionic phospholipid membranes mainly via two modes, one of which (termed Mode 2 here) is likely to be physiologically relevant. In this mode, the protein engages the membrane via its Ω1 loop and C-terminal helix, two structural regions that have been identified previously as being functionally important in StART domains (*Jentsch et al., 2018*; *Horenkamp et al., 2018*; *Gatta et al., 2018*; *Iaea et al., 2015*). Once membrane-bound, the protein adopts diverse conformations that are characterized by different extents of widening of the side-opening to the sterol-binding pocket. The side-opening in sterol-loaded Lam4S2 can be occluded by the polar side-chains of residues S181, D61, and K89, resulting in a low level of hydration within the cavity. In this condition, the cholesterol molecule is stably bound, with its hydroxyl group forming hydrogen-bonding interactions with residue Q121. Cholesterol egress is triggered stochastically, by gradual widening of the side-opening and concomitant penetration of water into the binding site. These dynamic events destabilize cholesterol in the binding site by ~4–5 kcal/mole, driving it from the binding site towards the membrane. The subsequent steps of the release process are enabled by repositioning of the Ω1 loop away from the C-terminal helix. This fully exposes the binding pocket to the membrane, that is widens the axial aperture, thus creating a continuous passageway to the membrane. The sequence of events by which sterol exits Lam4S2 and enters the membrane is shown in *Video 1*.

The overall process of cholesterol release requires that an energy barrier of ~6 kcal/mole is overcome. This value is in reasonable agreement with the ~10–15 kcal/mole estimate for the energy barrier for sterol extraction by Lam4S2 based on (i) measurement of its transport rate, (ii) the assumption that intermembrane sterol transfer is rate-limited by sterol pick-up/delivery, and (iii) the observation that the rate constant for this process can be described by a simple Arrhenius relationship (*Dittman and Menon, 2017*). Thus, using the measured transport rate of ~0.8 sterol molecules per second per Lam4S2 (*Jentsch et al., 2018*), and Arrhenius prefactors in the range $10^9$–$10^{10}$ s$^{-1}$ (*Dittman and Menon, 2017*), we estimate the energy barrier to be 20.9–23.2 $k_B T$, equivalent to ~12.5–14 kcal/mole. Overall, the computational findings reported here reveal that the

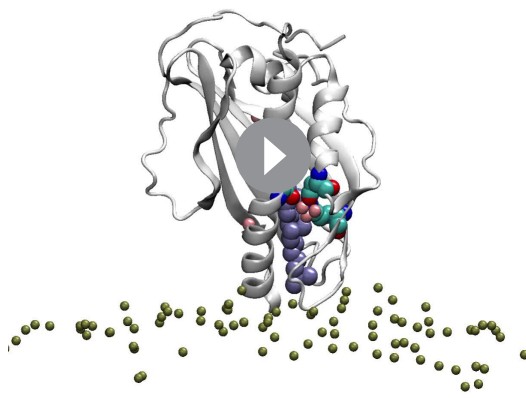

**Video 1.** Molecular dynamics trajectory of cholesterol egress from Lam4S2. The movie is based on one of the Stage 2 simulations of wild-type Lam4S2. The total length of the trajectory is 350 ns. In the movie, Lam4S2 is shown in white cartoon, the cholesterol molecule is represented in ice-blue colored space-fill, S181, D61, and K89 residues are drawn in space-fill, the oxygen atoms of water molecules in the sterol-binding site are depicted as pink spheres, the membrane leaflet to which Lam4S2 is bound is represented by the nearby lipid phosphate atoms (golden spheres), and lipid molecules within 3 Å of the cholesterol are shown in licorice representation. The rest of the simulation box is omitted. For clarity, the trajectory frames are smoothed for the movie.

https://elifesciences.org/articles/53444#video1

conformational state of the side-opening to the sterol-binding cavity in the Lam4S2 StARkin domain plays a major role in regulating the energetic stability of the sterol in the pocket.

This prediction was probed first computationally by analyzing MD trajectories of Lam4S2 in which residues that line the side-entrance to the binding site were substituted with alanine. For all three mutations (S181A, D61A and K89A), we found destabilization of cholesterol in the binding site. Using potential of mean force calculations, we found that the K89A mutation lowered the energy barrier for cholesterol release by ~2 fold compared with that for wild type Lam4S2. Experimental tests confirmed that the K89A mutant was non-functional, whereas the S181A mutant was only partially compromised in its ability to bind sterol, a defect that did not appear to influence its ability to rescue the nystatin-sensitivity of *lam2Δ* cells or to exchange sterols between membranes in vitro. Our test of the D61A mutant was limited to a cell-based assay in which it performed as well as the wild-type protein in rescuing the nystatin-sensitivity of *lam2Δ* cells.

Considering the functional importance of K89, and to a lesser extent S181, we examined the conservation of these residues in the Lam/GramD1 family using a previously reported structure-based sequence alignment (*Horenkamp et al., 2018*). We found that the positions aligning with K89 and S181 were among the residues with the highest conservation score. Interestingly, it has been noted that the side-chain of residue K910 in the S1 domain of Lam2 (Lam2S1), which aligns with K89 of Lam4S2

(note that K89 in Lam4S2 corresponds to K1031 in the full-length protein [*Table 1*]) is positioned slightly differently in the ergosterol-bound and *apo* structures (*Horenkamp et al., 2018*). This led to speculation that a path for ergosterol movement into and out of Lam2S1 could be enabled by movement of K910. Consistent with this, our study reveals that residue K89 in Lam4S2 does indeed reposition when cholesterol is released from the protein. Importantly, we find that this movement is a part of larger-scale dynamic changes involving neighboring polar residues, D61 and S181, that lead to widening of the side-opening to the binding pocket.

Our computational analysis points to the key role that solvation of the sterol-binding pocket plays in the process of cholesterol release. We find that water penetration destabilizes hydrogen-bonding interactions between the 3-β-OH of cholesterol and the side-chain of Q121, leading to initiation of sterol egress. Although the current computations have not directly addressed the mechanism of sterol *entry* into the binding site, the PMF profile that we report here suggests that the continuous water pathway connecting the binding site to the bulk solution, as observed in our simulations of *apo* Lam4S2 (*Figure 2—figure supplement 3*), should play an important role in the delivery of sterol into the binding site. In this respect, it is important to note that similar to Lam4S2 (*Figure 1—figure supplement 1B*), the sterol-binding pocket in all other Lam/GramD1 domains with known structure show strong polar characteristics. Furthermore, although in some X-ray structures of Lam/GramD1 StARkin domains, the polar head-group of the bound sterol is seen in direct contact with neighboring polar residues, in others it is engaged with the protein indirectly through water-mediated interactions. The former mode is observed in Lam4S2, Lam2S2 and GramD1a, whereas the latter mode is seen in Lam2S1. Interestingly, in both Lam4S2 and Lam2S2, the head-group of the bound sterol

**Table 1.** Constructs and strains.

**Bacterial plasmids (all constructs are in the pTrcHis6A expression vector and start with the sequence MGGSHHHHHHGMASHHHHHARALEVLFQGPM)**

| | |
|---|---|
| Lam4S2[1] | Lam4 946–1145 |
| Lam4S2(D61A)[3] | Lam4 946–1145 (D1003A) |
| Lam4S2(K89A)[3] | Lam4 946–1145 (K1031A) |
| Lam4S2(S181A)[3] | Lam4 946–1145 (S1123A) |
| Yeast plasmids | |
| GFP only[2] | pRS416 (CEN *URA3*): GFP + GFP |
| GFP-Lam4S2[2] | pRS416 (CEN *URA3*): GFP + Lam4 946–1155 + DV[4] |
| GFP-Lam4S2(D61A)[3] | pRS416 (CEN *URA3*): GFP + Lam4 946–1155 (D1003A) + DV[4] |
| GFP-Lam4S2(K89A)[3] | pRS416 (CEN *URA3*): GFP + Lam4 946–1155 (K1031A) + DV[4] |
| GFP-Lam4S2(S181A)[3] | pRS416 (CEN *URA3*): GFP + Lam4 946–1155 (S1123A) + DV[4] |
| Bacterial strain | |
| *E. cloni* EXPRESS BL21(DE3) | F– *ompT hsdSB* (rB– mB–) *gal dcm lon* λ(DE3 [*lacI lac*UV5-T7 gene one *ind1 sam7 nin5*]) |
| Yeast strain | |
| *lam2Δ* (also called *ysp2Δ*)[5] | *MATa his3Δ1 leu2Δ0 met15Δ0 ura3Δ0 ysp2Δ::hphNT1* |

[1] Described in *Jentsch et al. (2018)*.

[2] Described in *Gatta et al. (2015)*.

[3] Parentheses indicate point mutations, for example K89A, using the Lam4S2 numbering system of *Jentsch et al. (2018)*; residue numbering based on the entire Lam4 sequence is provided in the right-hand column.

[4] Two amino acids (DV) appended to the end of the Lam4S2 sequence.

[5] Described in *Roelants et al. (2018)*.

hydrogen-bonds to the side-chain of a Gln residue (Q121 in Lam4S2). In Lam2S1, on the other hand, the position aligning with Q121 is occupied by a small-size polar residue, Ser. Therefore, the head-group of the sterol does not form a direct hydrogen bond within the binding pocket of Lam2S1 but rather associates with the protein through water-mediated interactions. In GramD1a, in which the residue analogous to Q121 of Lam4S2 is also Ser, the bound sterol is seen in direct contact with another adjacent polar residue (Tyr). Similar to Lam/GramD1 StARkin domains, structural information on yeast Osh4/Kes1 (*Im et al., 2005*) in complex with different sterols reveals the prominence of water-mediated interactions between the 3-β-OH group of the bound sterol and a cluster of polar residues in the binding site. Furthermore, steered MD simulations of cholesterol release from Osh4 into water (*Singh et al., 2009*) suggested that the release process unfolds via a *molecular ladder* mechanism whereby the 3-β-OH group switches between different sets of water-mediated interactions with the binding-site residues as it leaves the protein. Taken together, the structural information highlights the importance of polar interactions for the stability of the sterol molecule in the binding site, consistent with our results demonstrating that disruption of these interactions by influx of water through the cavity side-opening leads to sterol release. Therefore, the molecular mechanism of sterol release that we have identified in Lam4S2 is likely to be generalizable to other homologous domains.

## Materials and methods

### Computational methods

Molecular constructs of wild-type Lam4S2

The computations were based on the X-ray structures of the second StARkin domain of Lam4, Lam4S2 (PDBIDs 6BYM and 6BYD) (*Jentsch et al., 2018*). In the 6BYM structure, Lam4S2 (residue sequence 4–196 in the numbering used in *Jentsch et al., 2018*, where residue 4 corresponds to Thr-946 in native Lam4) is in complex with 25-hydroxycholesterol, which is bound in the canonical sterol-binding pocket identified also in the StARkin domains of other Lam proteins (*Horenkamp et al.,*

*2018*; *Tong et al., 2018*). In the 6BYD model, Lam4S2 (residue sequence 4–200) is in the *apo* form. For the computational studies described here, the oxysterol in the 6BYM structure was replaced by cholesterol and the molecular models of Lam4S2 in both 6BYM and 6BYD structures were completed using modeller 9v1 (RRID:SCR_008395) (*Eswar et al., 2006*) to add respective missing residue stretches, that is residues 1–3 and 197–203 to the 6BYM structure, and residues 1–3 and 201–203 to the 6BYD structure.

## Unbiased MD simulations of sterol-bound wild-type Lam4S2

An all-atom model lipid membrane with the composition of 'Acceptor' liposomes in sterol transport assays (*Jentsch et al., 2018*) was prepared using the CHARMM-GUI web server (RRID:SCR_014892) (*Jo et al., 2009*). Thus, symmetric lipid bilayer containing 70% DOPC, 15% PI, 10% DOPE, and 5% DOPS (400 lipids in total on the two leaflets) was assembled, solvated (using water/lipid number ratio of 50) and ionized with 0.1M K$^+$Cl$^-$ salt. This system was subjected to MD simulations for 30 ns using NAMD version 2.13 (RRID:SCR_014894) (*Phillips et al., 2005*) and the standard multi-step equilibration protocol provided by CHARMM-GUI.

After this equilibration phase, the bilayer system was stripped of all water molecules and solution ions and the cholesterol-bound Lam4S2 domain (6BYM) was placed near the membrane surface so that the distance between any atom of the protein and any atom of the lipid molecules was $\geq$10 Å (see *Figure 1B*). The protein-membrane complex was solvated (using water/lipid number ratio of ~145) and ionized (with 0.1M K$^+$Cl$^-$ salt). The resulting system contained ~234,000 atoms in total.

The Lam4S2–membrane complex was equilibrated using a multi-step protocol (*Shi et al., 2008*) during which the backbone of the protein was first harmonically constrained and subsequently gradually released in three steps of 5 ns each, changing the restraining force constants from 1, to 0.5, and 0.1 kcal/ (mol Å$^2$), respectively. This step was followed by 6-ns-long unbiased MD simulations carried out using the NAMD 2.13 package. After this short run, the velocities of all the atoms were reset and the system was simulated with ACEMD software (*Harvey et al., 2009*) in 10 statistically independent replicates (Stage 1 ensemble simulations), each for 320 ns, resulting in a cumulative time of 3.2 μs for Stage 1 runs.

As described in 'Results', Stage 1 simulations sampled events of spontaneous binding of Lam4S2 to the membrane. We randomly selected 10 frames from Stage 1 trajectories in which Lam4S2 was seen to be interacting with the lipid bilayer as in *Figure 1E*, and initiated a new set of simulations with ACEMD (Stage 2 ensemble simulations) in which the 10 chosen structures were run in 10 statistically independent replicates each (i.e. 100 independent simulations). Each of the 100 copies were simulated for 375 ns, resulting in a cumulative time of 37.5 μs for Stage 2 runs.

## Unbiased MD simulations of *apo* wild-type Lam4S2

Simulations of the *apo* wild-type Lam4S2 protein (6BYD) followed the same protocol as described above for Stage 1 simulations of sterol-bound Lam4S2, with the only difference being the lipid membrane composition. Thus, in the manner identical to the sterol-bound Lam4S2, the *apo* protein was placed near the surface of the all-atom model lipid membrane (assembled with CHARMM-GUI) with the composition of 'Donor' liposomes in sterol transport assays (*Jentsch et al., 2018*). This symmetric bilayer contained 31% DOPC, 23% DOPE, 23% DOPS, and 23% cholesterol (400 lipids in total on the two leaflets). As the purpose of these simulations was to quantify solvation of the empty sterol-binding site, this system was only considered for Stage 1 simulations (cumulative time of 3.2 μs) and was not subjected to a subsequent (Stage 2) phase.

## Unbiased MD simulations of the mutant Lam4S2 systems

Using the FoldX server (RRID:SCR_008522) (*Schymkowitz et al., 2005*), three single mutations, K89A, S181A, and D79A in Lam4S2 were introduced into two separate frames of one of the Stage two ensemble trajectories of the wild type protein system (see 'Results'). The resulting structures (two per mutant) were energy-minimized for 100 steps and then simulated in five independent replicates each for 150 ns using ACEMD. This resulted in 10 statistically independent MD trajectories per mutant, totaling 1.5 μs.

## Parameters and force-field for MD simulations

All the simulations performed with NAMD 2.13 implemented the *all* option for rigidbonds, 2fs integration time-step, and PME for electrostatics interactions (*Essmann et al., 1995*), and were carried out in NPT ensemble under semi-isotropic pressure coupling conditions, at a temperature of 310 K. The Nose-Hoover Langevin piston algorithm (*Phillips et al., 2005*) was used to control the target p=1 atm pressure with the LangevinPistonPeriod set to 100 fs and LangevinPistonDecay set to 50 fs. The van der Waals interactions were calculated by applying a cutoff distance of 12 Å and switching the potential from 10 Å. In addition, the vdwforceswitching option was set to on.

The simulations carried out with ACEMD software implemented the PME method for electrostatic calculations, and were carried out according to the protocol developed at Acellera and implemented by us previously (*Harvey et al., 2009*; *Khelashvili et al., 2015*),with four fs integration time-step and the standard mass repartitioning procedure for hydrogen atoms. The computations were conducted under the NVT ensemble (at T = 310 K), using the Langevin Thermostat with Langevin Damping Factor set to 0.1.

For all the simulations, the CHARMM36 force field parameters were used for proteins, lipids, sterols, and ions (*Phillips et al., 2005*; *Lee et al., 2016*).

## Umbrella sampling MD simulations of wild-type and K89A Lam4S2

Biased MD simulations of cholesterol release from the wild-type and the K89A mutant Lam4S2 were performed using the umbrella sampling approach. The position of the translocated cholesterol was restrained to different locations along the translocation pathway (see 'Results') using as a collective variable the z-directional distance ($d_{Z(chol-121)}$ along the axis perpendicular to the membrane plane), between the cholesterol oxygen and the $C_\alpha$ atom of residue Q121 (see *Figure 1A*). 19 windows spaced 1Å apart in the range of $d_{Z(chol-121)} \in$ [2Å; 20Å] were considered, and the dynamics of the sterol molecule in each window was restrained by applying a force constant of 2.5 kcal/mol $\cdot$ Å$^2$. The rest of the parameters for the umbrella sampling runs were as follows: *width* – 2Å, and both *lowerwallconstant* and *upperwallconstant* set to 25 kcal/mol $\cdot$ Å$^2$. Each umbrella window was simulated for 50 ns, which resulted in good overlap between adjacent windows (*Figure 4—figure supplement 3A,B*).

The potential of mean force (PMF) along the collective variable was constructed with the Weighted Histogram Analysis Method (WHAM) (Version 2.0.9; *Grossfield, 2013*). For the WHAM calculations, only the last 25 ns trajectory segments of each umbrella window were used. The tolerance parameter was set to 0.0001. To estimate error bars on the PMF, for each umbrella window, the first decorrelation time was calculated as a time-constant from a single exponential fit to the auto-correlation vs time data (*Figure 4—figure supplement 3C,D*). The error bars were then constructed with Monte Carlo bootstrapping error analysis in the WHAM software on the decorrelated data points using *num_MC_trials* of 1000.

Umbrella sampling MD simulations were also carried out on wild-type Lam4S2 in solution. To this end, structures of the protein–cholesterol complexes from the same 19 windows were extracted and immersed into a water box and the dynamics of the system was studied by restraining the sterol molecule in each window as described above. To prevent the protein–cholesterol complex from rotating in solution, we applied an additional harmonic bias (force-constant of 4000 kcal/mol $\cdot$ Å$^2$) on the protein backbone around the identity rotation matrix (1.0, 0.0, 0.0, 0.0) using the colvar 'orientation' module in NAMD 2.13. We note that this rotational restraint does not influence internal degrees of freedom but rather ensures that in each window, the protein–cholesterol complex remains aligned along the z axis throughout the umbrella sampling protocol.

## Dimensionality reduction using the time-structure-based independent component analysis (tICA)

To facilitate analysis of the process of cholesterol release from the Lam4S2 domain in the MD simulations, we performed dimensionality reduction using the tICA approach (*Molgedey and Schuster, 1994*; *Naritomi and Fuchigami, 2011*; *Pérez-Hernández et al., 2013*; *Schwantes and Pande, 2013*) as described previously (*Morra et al., 2018*; *Lee et al., 2018*; *Razavi et al., 2018*). To define the tICA space, we used several dynamic variables extracted from the analysis of the ensemble MD trajectories that quantify the dynamics of the cholesterol, the extent of exposure of the sterol

binding site to the solvent, and the dynamics of the functionally important Ω1 loop. These variables include (see *Figure 1A*): (1)-the minimum distance between the hydroxyl oxygen atom of the translocated cholesterol and residue Q121 ($d_{chol-121}$); (2)-the root-mean-square deviation (RMSD) of the cholesterol molecule from its position in the binding site; (3)- distance between the hydroxyl oxygen of S181 and $C_\gamma$ carbon of D61 ($d_{61-181}$); (4)-$C_\alpha - C_\alpha$ distance between residue I95 in the $\Omega_1$ loop and residue A169 in the C-terminal helix ($d_{95-169}$); (5)-number of water molecules in the interior of the protein (defined as the number of water oxygens found within 5 Å of the side-chains of the following protein residues – 189, 185, 181, 154, 152, 136, 138, 140, 142, 123, 121, 119, 117, 102, 104, 106, 108, but farther than 5 Å from the following residues – 116, 118, 109, 86, 103, 105); (6)-the number of lipid phosphate atoms with 3 Å of the translocated cholesterol molecule.

Using these six CVs as components of the data vector $X$, the slowest reaction coordinates of a system were found as described previously (*Morra et al., 2018*; *Razavi et al., 2018*; *Razavi et al., 2017*), by constructing a time-lagged covariance matrix (TLCM): $C_{TL}(\tau)=<X(t)X^T(t+\tau)>$ and the covariance matrix $C=<X(t)X^T(t)>$, where $X(t)$ is the data vector at time $t$, $\tau$ is the lag-time of the TLCM, and the symbol $<...>$ denotes the time average. The slowest reaction coordinates are then identified by solving the generalized eigenvalue problem: $CTLV = CV\Lambda$, where $\Lambda$ and $V$ are the eigenvalue and eigenvector matrices, respectively. The eigenvectors corresponding to the largest eigenvalues define the slowest reaction coordinates.

### Calculation of electrostatic potential

The electrostatic potential for the wild-type and the K163D/K167D double mutant Lam4S2 constructs was calculated by solving the non-linear Poisson-Boltzmann (NLPB) equation with APBS software (*Baker et al., 2001*) as described before (*Khelashvili, 2019*). The protein and solvent dielectric constants were set to 2 and 78.54, respectively. The solution contained 150 mM of monovalent mobile ions. The NLPB equation was solved on a 3D cubic grid of 256 Å$^3$ volume discretized into 1 Å$^3$ grid elements, using the multigrid method (mg-manual) and multipole boundary conditions.

### Experimental methods
#### Lam4S2 mutants
Point mutants of Lam4S2 (D61A, K89A and S181A) were generated by PCR mutagenesis and confirmed by sequencing. The constructs and PCR primers are detailed in *Table 1* and *Table 2*.

### Protein expression and purification

Lam4S2 and point mutants were expressed in *E. coli* as His-tagged proteins (*Table 1*), and purified by affinity chromatography on Ni-NTA resin, followed by size exclusion chromatography (SEC) using a Superdex 200 Increase 15/300 GL column. The purification procedure was as previously described (*Jentsch et al., 2018*), except that the proteolysis step to remove the affinity tag was omitted and SEC was carried out in 20 mM HEPES [pH 7.5] and 150 mM NaCl. The purified protein was snap frozen in small aliquots and stored at −80˚C. Prior to use, aliquots were thawed and subjected to brief microcentrifugation to remove any aggregated material. Purified proteins were quantified by absorbance at 280 nm; quality control included analysis by circular dichroism (CD) as described previously (*Jentsch et al., 2018*), and re-analysis by SEC using the buffer conditions described above.

### Sterol transport assay

The assay (illustrated in *Figure 7A*) was performed and analyzed as previously described (*Jentsch et al., 2018*; *Chauhan et al., 2019*) using anionic donor and acceptor liposomes (donor lipid composition: DOPC, DOPE, DOPS, DHE [31, 23, 23, and 23 mol %, respectively]; acceptor lipid

**Table 2.** Primers used for mutagenesis.

| Mutation | Forward primer | Reverse primer |
|---|---|---|
| D61A | CAGAAAGTTATCACTAGAGCTAAGAATAATGTCAATGTGG | CCACATTGACATTATTCTTAGCTCTAGTGATAACTTTCTG |
| K89A | CACTATGAGTACACGGCGAAATTGAACAATTCTATC | GATAGAATTGTTCAATTTCGCCGTGTACTCATAGTG |
| S181A | GAGGGTCAGAAGGTTGCTGTCGATTACATGCTA | TAGCATGTAATCGACAGCAACCTTCTGACCCTC |

composition: DOPC, DOPE, liver PI, DOPS, dansyl-PE [70, 7, 15, 5, and 3 mol %, respectively]). Briefly, assays were carried out at 23˚C in a quartz cuvette with constant stirring using a temperature-controlled Horiba Fluoromax Plus-C spectrofluorometer. The total sample volume was 2 ml, with 0.1 mM each of donor and acceptor liposomes (final concentration, based on measurement of inorganic phosphate after acid hydrolysis of the vesicles) and 0.05 µM or 0.1 µM Lam4S2 (final concentration) in 20 mM PIPES (pH 6.8), 3 mM KCl and 10 mM NaCl (assay buffer). Fluorescence was monitored for ~2500 s using $\lambda_{ex}$ = 310 nm and $\lambda_{em}$ = 525 nm and a data acquisition frequency of 1 Hz. Acceptor liposomes were added to donor liposomes in the cuvette, and after 60 s, 200 µl of Lam4S2 (or Lam4S2-mutant), diluted as needed in assay buffer, was added. For control assays, 200 µl of assay buffer was added. All traces were offset-corrected such that the fluorescence signal and time at the point of Lam4S2 (or buffer) addition were each set to zero. The maximum possible FRET signal was determined from assays using 0.1 µM wild-type Lam4S2, where the fluorescence readout reached a plateau value within 2000 s. Traces from such assays (done in replicate) were fit to a mono-exponential function, and the plateau value obtained ($FRET_{max}$) was used to constrain the mono-exponential fits of all other traces. Traces from different assays were compiled after data fitting by setting $FRET_{max}$ = 1.

## Sterol-extraction assay

Sucrose-loaded liposomes were prepared as follows. Lipids (2 µmol total, of a mixture of DOPC, DOPE, DOPS, cholesterol (49, 23, 23, 5 mol %, respectively, containing a trace amount of [³H]cholesterol and N-rhodamine-DHPE) were dried in a glass screw-cap tube under a stream of nitrogen, then resuspended in 1 ml assay buffer (20 mM PIPES [pH 6.8], 3 mM KCl, and 10 mM NaCl) supplemented with 250 mM sucrose, by agitating on a Vibrax orbital shaker for 30 min at 1200 rpm. The resulting suspension was subjected to five cycles of freeze–thaw (immersion in liquid nitrogen, followed by thawing at room temperature), before being extruded 11 times through a 200 nm membrane filter using the Avanti Mini-Extruder. After extrusion, extravesicular sucrose was removed by diluting the vesicles 4x in assay buffer and centrifuging in a Beckman TLA100.3 rotor (75,000 rpm, 1 hr, 4˚C). The supernatant was carefully removed from the pelleted vesicles (easily discernable because of their pink color due to rhodamine-DHPE), before resuspending the vesicles in 1 ml of assay buffer. Aliquots of the sample (5 µl) were removed at different points of preparation (after the freeze–thaw step, post-extrusion and after final resuspension) and taken for liquid scintillation counting to track lipid recovery by monitoring [³H]cholesterol.

The ability of Lam4S2 (wild-type and point mutants) to extract cholesterol from the vesicles was determined as follows. Liposomes (15 µl, ~1500 pmol cholesterol) and protein (500–750 pmol, as indicated) were combined in assay buffer to a total volume of 500 µl. The mixture was incubated at room temperature for 1 hr, before removing a 20 µl aliquot for liquid scintillation counting. The remainder of the sample was centrifuged in a Beckman TLA100.2 rotor (75,000 rpm, 1 hr, 4˚C). Most of the supernatant (350 µl) was transferred to a fresh 1.5 ml tube, whereas the remainder was removed immediately and discarded. The pellet was resuspended in 100 µl assay buffer containing 5% (w/v) SDS. Duplicate aliquots (50 µl each) of the supernatant were taken for liquid scintillation counting to determine the amount of extracted cholesterol. Protein in the remainder of the supernatant (250 µl) and the resuspended pellet was precipitated by adding 1.2 ml ice-cold acetone, followed by overnight incubation at −20˚C. The precipitated proteins were pelleted by centrifugation, air-dried after removal of the acetone, and dissolved in SDS gel loading buffer. The relative amount of protein in the supernatant and pellet fractions was determined by SDS-PAGE, Coomassie staining and quantification of band intensity using Image J software (RRID:SCR_003070). The data are represented as pmol cholesterol extracted/pmol protein in the supernatant.

## Acknowledgements

We thank Tim Levine and Ganiyu Alli-Balogun (Department of Cell Biology, University College London Institute of Ophthalmology) for plasmids, Trudy Ramlall (Eliezer laboratory, Weill Cornell Medical College) for help with protein purification, and Harel Weinstein for insightful comments on the manuscript. GK would like to thank Jerome Henin for his advice on umbrella sampling simulations. AKM would like to acknowledge Bob Dylan and Sam Canis for the usual reasons, as well as the unusual stimulation provided by Typhoon Hagibis. The computational work was performed using

resources of the Oak Ridge Leadership Computing Facility (ALCC allocation BIP109 and Director's Discretionary allocation) at the Oak Ridge National Laboratory, which is supported by the Office of Science of the U.S. Department of Energy under contract no. DE-AC05-00OR22725, and the computational resources of the David A Cofrin Center for Biomedical Information in the HRH Prince Alwaleed Bin Talal Bin Abdulaziz Alsaud Institute for Computational Biomedicine at Weill Cornell Medical College.

## Additional information

### Funding

| Funder | Grant reference number | Author |
|---|---|---|
| National Institutes of Health | R37AG019391 | David Eliezer |
| 1923 Fund | | George Khelashvili |

The funders had no role in study design, data collection and interpretation, or the decision to submit the work for publication.

### Author contributions

George Khelashvili, Conceptualization, Formal analysis, Investigation, Methodology, Writing—original draft, Writing—review and editing; Neha Chauhan, Kalpana Pandey, Formal analysis,Investigation, Methodology, Writing—review and editing; David Eliezer, Conceptualization,Writing—review and editing; Anant K Menon, Conceptualization, Formal analysis,Methodology, Supervision, Project administration, Writing—original draft, Writing—review and editing

### Author ORCIDs

George Khelashvili https://orcid.org/0000-0001-7235-8579
Neha Chauhan https://orcid.org/0000-0003-1497-3359
David Eliezer https://orcid.org/0000-0002-1311-7537
Anant K Menon https://orcid.org/0000-0001-6924-2698

### Decision letter and Author response

Decision letter https://doi.org/10.7554/eLife.53444.sa1
Author response https://doi.org/10.7554/eLife.53444.sa2

## Additional files

### Supplementary files

• Transparent reporting form

### Data availability

All data generated or analysed during this study are included in the manuscript and supporting files. Source data files have been provided for Figures 1, 2 and 4.

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
