## [Decision Letter]

[Editors’ note: a previous version of this study was rejected after peer review, but the authors submitted for reconsideration. The first decision letter after peer review is shown below.]

Thank you for submitting your work entitled "Water for sterol: an unusual mechanism of sterol egress from a StARkin domain" for consideration by *eLife*. Your article has been reviewed by three peer reviewers, one of whom is a member of our Board of Reviewing Editors, and the evaluation has been overseen by a Senior Editor. The reviewers have opted to remain anonymous.

Our decision has been reached after consultation between the reviewers. Based on these discussions and the individual reviews below, we regret to inform you that your work will not be considered further for publication in *eLife*.

Your study examines the interesting and poorly-understood question of how a sterol-binding protein releases its bound sterol into a lipid bilayer. You provide several lines of evidence that the bound sterol is ejected from a sterol-binding STARkin protein domain by the penetration of water into the binding pocket. Your finding that the binding pocket accommodates the hydrophobic sterol as well as water is surprising and intriguing.

All the reviewers agreed that this study was thought-provoking and constituted a significant Research Advance. However, they also raised several concerns, addressing which would take more than the two-month revision window. The reviewers welcome assessment of a revised submission, but only if it addressed all the points raised in the full reviews listed below.

*Reviewer #1:*

This study by Khelashvili et al. follows up on two previous studies, one in *eLife* where ER proteins with sterol-binding StARkin domains were identified (2015) and a second in JBC where the structure of a yeast StARkin domain was determined (2018). Those previous studies did not yield mechanistic insights into how the StARkin domain delivered its bound sterol to the membrane. Here, the authors use molecular dynamics simulations to show that the ejection of the sterol into the membrane is preceded by penetration of water into the binding cavity – hence the "water for sterol" title. Since sterols are extremely hydrophobic, this is a surprising finding in that the binding pocket seems to accommodate the hydrophobic sterol as well as the complete opposite molecule on the hydrophobicity scale – water! The simulations identify a lysine residue (K89) that is important for modulating this egress step. This is an intriguing paper that could constitute an advance, but there are two main issues that need to be addressed.

1) This study focuses on the transfer of cholesterol from cholesterol-Lam4S2 complex to the membrane. The crystal structure in the Jentsch et al. paper shows sterol-bound Lam4S2, but the bound sterol is 25-hydroxycholesterol (25HC), not cholesterol (or ergosterol). The complex that is the starting point here replaces the bound 25HC with cholesterol. To fortify their starting point, the authors should show that bound 3H-25HC is competed away by unlabeled 25HC as well as cholesterol (or ergosterol). The authors report a sterol extraction assay with 3H-cholesterol in Figure 6D, so a binding assay seems possible to design to address this point. Alternatively, the simulations could be done with Lam4S2-ergosterol complexes, which may yield more robust results since it is the natural ligand, and is likely the sterol that is transported. I understand the authors want to make a general point about these StARkin domains, but the use of ergosterol instead of cholesterol would not dampen the provocative "water-for-sterol" ejection mechanism.

2) The elegant simulations indicate that the first step in sterol egress is widening of the side-entrance (microstate 2 and 3) to allow more water molecules into the binding pocket. Presumably, this only happens when the Lam4S2/sterol binds to or is close to a membrane? What in the membrane-bound state mediates this widening? In other words, why doesn't the widening and water penetration occur when the Lam4S2 is in solution?

*Reviewer #2:*

In their manuscript Menon and co-workers examine at the atomistic level how cholesterol exits the binding pocket of the Starkin domain of Lam4 (LAMS2), a recently discovered sterol transfer protein that functions in ER/PM contacts in yeast. By an impressive number of unbiaised MD simulations and elegant analyses by PCA, they reveal a mechanism by which water molecules enter via a fracture along the sterol-binding pocket to destabilize a key interaction between the 3-OH of sterol and Lam4S2. A triad of residues closing the top of this fracture is involved in the pace of this mechanism and plays a substantial role in stabilizing sterol in Lam4. Overall the work is very interesting mostly due to the in silico approaches. Indeed we see sterol egress in a non-steered MD and its entry in a bilayer. Yet it is difficult in the second-part of the manuscript to apprehend to what degree the water-mediated mechanism of sterol exit is supported by in vivo/in vitro experiments. Rather, experimental data suggest that a point mutation of a residue K89, found in simulation to limit sterol stability, renders Lam4 inactive but is this sufficient to validate the occurrence of the water-mediated mechanism? I think the manuscript deserves to be published in *eLife* but with more experimental data to buttress the MD simulations.

- Stage 1 simulations identify two poses for LamS2 on a bilayer: one seems impossible due to steric constraints imposed by the N and C regions around the Starkin domain whilst the second one seems compatible. Functional assays suggest that a few residues found to insert in the membrane by MD are indeed key for membrane recognition. However, this does not tell that the selected orientation of Lam4S2 in MD is the genuine one. Authors should provide evidences on this as this orientation is the starting point for the analysis of sterol exit, thus of the rest of the study.

- There is somehow a gap between MD and functional assays as there is no structural analysis in between. Indeed, there is no direct experiments to measure the accessibility of the binding pocket to water molecules in the presence/absence of the ligand and/or mutation – possibly, H-D exchange experiments by NMR using D_2_O could be useful in that respect. Also there is no assay to quantitatively measure the stability of sterol in Lam4S2 at equilibrium and/or its release in membrane. Osh3 and Osh4 were found to load DHE by measuring energy transfer between W residues, surrounding the pocket, and DHE (Tong et al., Structure. 2013, 21(7):1203-13; De Saint-Jean et a, J Cell Biol. 2011;195(6):965-78.). Offloading of DHE into membrane can be measured also by FRET in lipidic vesicles in real time. Authors should try such approaches that could more directly support MD simulations. Last they might use transport rates inferred from DHE transport assays with Lam4S2 to estimate an energy barrier for sterol extraction by the protein instead of quoting Dittman and Menon, 2017.

- Authors should check at least that single-mutation K89A or S181A does not affect the membrane-binding ability of Lam4S2

- it is quite surprising that 2 mutants out of 3 are not defective in yeast. In particular the D61A mutant do not phenocopy the K89A mutant, which is unexpected considering the existence of a electrostatic interaction with K89 and its central position in the triad of residues, making also contact with S181. Is there any explanation for this ?

*Reviewer #3:*

The mechanism of cholesterol release from endoplasmic reticulum domains that possess sterol-binding domains was studied. A computer model of the protein was derived from recently published crystallographic and spectroscopic structural data. Simulations have been well executed. Appropriate literature regarding previous evidence has been cited. The mildly steered molecular simulations provide deeper insight into the process of cholesterol release as well as its energetics. It highlights the role of water molecules in release of cholesterol from the binding domain. Conclusions are strengthened by studying the influence of alanine mutations on cholesterol binding and release both experimentally and in simulations. The paper is informative and very well written.

I suggest addressing the following question in more detail: Sterol molecules are amphipathic. The binding pocket for sterols must accommodate both polar and hydrophobic regions of cholesterol. How does the binding pocket deal with water influx into hydrophobic areas of the binding pocket? Is there evidence for structural changes in the protein to reduce energetically unfavorable interactions?

---

## [Author Response]

[Editors’ note: the author responses to the first round of peer review follow.]

All the reviewers agreed that this study was thought-provoking and constituted a significant Research Advance. However, they also raised several concerns, addressing which would take more than the two-month revision window. The reviewers welcome assessment of as revised submission, but only if it addressed all the points raised in the full reviews listed below.Reviewer #1:[…] This is an intriguing paper that could constitute an advance, but there are two main issues that need to be addressed.1) This study focuses on the transfer of cholesterol from cholesterol-Lam4S2 complex to the membrane. The crystal structure in the Jentsch et al. paper shows sterol-bound Lam4S2, but the bound sterol is 25-hydroxycholesterol (25HC), not cholesterol (or ergosterol). The complex that is the starting point here replaces the bound 25HC with cholesterol. To fortify their starting point, the authors should show that bound 3H-25HC is competed away by unlabeled 25HC as well as cholesterol (or ergosterol). The authors report a sterol extraction assay with 3H-cholesterol in Figure 6D, so a binding assay seems possible to design to address this point.

Lam4S2 has been previously shown to bind cholesterol, dehydroergosterol (DHE) and 25HC; binding to ergosterol was shown indirectly via competition experiments.

Thus, in Gatta et al., 2015, on which this Research Advance is based, we showed that Lam4S2 binds DHE and cholesterol. DHE binding was demonstrated using a FRET assay, and the extent of binding was reduced in the presence of cholesterol or ergosterol. Binding of Lam4S2 to cholesterol was also explicitly shown by demonstrating the specific ability of the protein to extract radiolabeled cholesterol from permeabilized HL-60 cells that had been metabolically radiolabeled with [^14^C]acetate to label all lipids. As noted by the reviewer, our crystal structure (Jentsch et al., 2018) additionally shows that the protein binds 25HC. We now report competition data that confirm these findings. Using the FRET assay of Gatta et al., 2015, we show that DHE binding is reduced by ergosterol and also by 25HC (Figure 1—figure supplement 2B). It is also noteworthy that binding of either 25HC ((Jentsch et al., 2018) or cholesterol (Gatta et al., 2018) to Lam4S2 results in similar changes to the NMR spectrum of the protein, indicating that the two sterols bind in a highly similar fashion.

These cumulative data – indicating equivalence of binding of all four sterols – are supported by structural models (Figure 1A (Lam4S2-cholesterol), Figure 1—figure supplement 2C (Lam4S2-ergosterol), Figure 1—figure supplement 2C (Lam4S2-25HC)), which show that 25HC, ergosterol and cholesterol adopt the same pose within the binding site. We therefore believe that our studies with cholesterol-bound Lam4S2 provide insight into the ejection mechanism for any of the four sterols mentioned here. See also our last response to point 1 below.

We have surveyed these points at the beginning of the Results section to clarify our choice of cholesterol-bound Lam4S2 for the simulations.

Alternatively, the simulations could be done with Lam4S2-ergosterol complexes, which may yield more robust results since it is the natural ligand, and is likely the sterol that is transported. I understand the authors want to make a general point about these StARkin domains, but the use of ergosterol instead of cholesterol would not dampen the provocative "water-for-sterol" ejection mechanism.

As per the reviewer's suggestion, we built the ergosterol-Lam4S2 complex *in silico* and studied its dynamics. To this end we docked ergosterol into Lam4S2, using the cholesterol-Lam4S2 structure which was used as the starting point for the Stage 2 ensemble simulations that we report. The resulting ergosterol-Lam4S2 system was then subjected to MD simulations using NAMD 2.13. The model was first energy-minimized and then equilibrated with unbiased MD for ~23 ns. In Figure 1—figure supplement 2C we show the structure of the complex in the final frame of the MD trajectory. As can be seen from the figure, ergosterol-bound Lam4S2 is structurally similar to cholesterol-bound Lam4S2 from the ensemble of states representing Microstate 1. Although significantly longer and computationally expensive MD trajectories will need to be accumulated in order to simulate the full process of ergosterol release from the binding site, these results strongly indicate that the release mechanism for ergosterol will be the same as the one we describe for cholesterol.

2) The elegant simulations indicate that the first step in sterol egress is widening of the side-entrance (microstate 2 and 3) to allow more water molecules into the binding pocket. Presumably, this only happens when the Lam4S2/sterol binds to or is close to a membrane? What in the membrane-bound state mediates this widening? In other words, why doesn't the widening and water penetration occur when the Lam4S2 is in solution?

To address the questions posed, we quantified the energetics of cholesterol release from Lam4S2 in water, in the same way as we did for Lam4S2 in the presence of the membrane. Specifically, we calculated the potential of mean force (PMF) as a function of d_z(chol-121)_ (z-directional distance between the cholesterol oxygen and C_α_ of residue Q121) from umbrella sampling simulations in water. The resulting PMF (black trace, Figure 4—figure supplement 2A) is clearly different from the PMF that we calculated in the presence of the membrane (red trace, Figure 4—figure supplement 2A). Thus, we find that the energy difference between the bound and released states in water is >10 kcal/mole which is a ~2-fold increase compared to the energy barrier for sterol release into the membrane. Furthermore, the PMF in water no longer has the secondary minimum (at ~12-14Å) that we observed in the PMF in the presence of the membrane. These major differences stem from the fact that during the egress process in the presence of the membrane, exposure of the hydrophobic tail of cholesterol to solvent is limited while this is clearly *not* the case when the sterol comes out into water (Figure 4—figure supplement 2B, left panels). Indeed, during egress from the Lam4S2 binding pocket in the presence of the membrane (as described in the original manuscript) the cholesterol tail becomes gradually encapsulated by the hydrophobic chains of neighboring lipids as it fully embeds into the lipid bilayer. In contrast, in water, cholesterol is released straight into the solvent, a process that is less energetically favorable as might be expected. Interestingly, the dynamics of the Ω_1_ loop (quantified by d_65_-d_169_ distance in Figure 4—figure supplement 2B, right panels) appear to be very similar in the two environments. These new computational results are included in the revised manuscript (Figure 4—figure supplement 2 and in the second paragraph of the subsection “K89A-Lam4S2 has a lower energy barrier for cholesterol release”).

Reviewer #2:[…] I think the manuscript deserves to be published in eLife but with more experimental data to buttress the MD simulations.

We thank the reviewer for the overall positive evaluation of our manuscript and for recognizing the value and quality of the computational studies presented in the paper. The revised manuscript contains substantial additional data, and we believe that these data together with our response to the specific points raised by all reviewers addresses the link between MD predictions and experimental validation.

- Stage 1 simulations identify two poses for LamS2 on a bilayer: one seems impossible due to steric constraints imposed by the N and C regions around the Starkin domain whilst the second one seems compatible. Functional assays suggest that a few residues found to insert in the membrane by MD are indeed key for membrane recognition. However, this does not tell that the selected orientation of Lam4S2 in MD is the genuine one. Authors should provide evidences on this as this orientation is the starting point for the analysis of sterol exit, thus of the rest of the study.

We assume that by referring to the selected binding mode as “genuine” the reviewer questions whether Mode 2 identified from our computations is functionally important. The standard approach to establish the relevance of computational results is to use them to make testable predictions, and we have done this. Thus, we have identified residues (K163 and K167) that are important for Mode 2 binding (now illustrated in Figure 1—figure supplement 3A, B). Indeed, we showed previously that charge reversal mutation of these residues perturbs function, as the K163D/K167D double mutant is unable to transport sterol between vesicles (Jentsch et al., 2018). The reviewer suggests that these mutations may simply prevent membrane recognition/binding, and therefore do not speak specifically to the significance of Mode 2. To explore this further, we obtained new data (Figure 1—figure supplement 3C) showing that the K163D/K167D mutation does not eliminate membrane recognition as suggested by the reviewer. Instead, this mutant binds to membranes as well as the wild type protein. New simulations (Figure 1—figure supplement 3D/E) reveal that this mutant destabilizes Mode 2 binding while retaining Mode 1 binding. Given the loss-of-function phenotype of this double mutant, these new results support the physiological relevance of Mode 2 as well as our assertion that Mode 1 represents an artifactual and non-functional binding mode. With that, our new results provide additional consistency between simulation and experiments. An updated discussion of these issues is provided in the subsection “Lam4S2 associates with the membrane via its Ω1 loop and C-terminal helix”.

- There is somehow a gap between MD and functional assays as there is no structural analysis in between. Indeed, there is no direct experiments to measure the accessibility of the binding pocket to water molecules in the presence/absence of the ligand and/or mutation – possibly, H-D exchange experiments by NMR using D2O could be useful in that respect.

The reviewer suggests H-D exchange to measure the accessibility of the binding pocket to water in the presence/absence of the ligand. It is not entirely clear what one might hope to learn from such measurements. Clearly, in the absence of ligand, the binding pocket is accessible to water (see our published apo crystal structure (Jentsch et al., 2018) as well as the water count in apo Lam4S2 simulations presented in Figure 2—figure supplement 3) and one would see significant H-D exchange. In the presence of ligand, the pocket would be protected to some extent from such exchange, but these observations would not fill the gap referred to by the reviewer as we have already observed the endpoints experimentally in crystal structures (apo protein and sterol-loaded protein). The requested experiments would most likely provide information that is redundant with the available structural information without lending new insights. To truly fill the gap between simulations and functional assays, one would need to monitor H-D exchange during cholesterol loading or unloading, and to correlate changes in exchange rates with the departure of sterol, but this would immensely difficult, if not outright impossible. Indeed, the strength and purpose of the MD simulations is to provide exactly this type of information since it is not readily obtainable experimentally.

Also there is no assay to quantitatively measure the stability of sterol in Lam4S2 at equilibrium and/or its release in membrane. Osh3 and Osh4 were found to load DHE by measuring energy transfer between W residues, surrounding the pocket, and DHE (Tong et al., Structure. 2013, 21(7):1203-13; De Saint-Jean et a, J Cell Biol. 2011;195(6):965-78.). Offloading of DHE into membrane can be measured also by FRET in lipidic vesicles in real time. Authors should try such approaches that could more directly support MD simulations. Last they might use transport rates inferred from DHE transport assays with Lam4S2 to estimate an energy barrier for sterol extraction by the protein instead of quoting Dittman and Menon, 2017.

We make several points in our response to these comments from the reviewer.

First, we do indeed provide data on the 'stability' of sterol in Lam4S2. Using a standard assay in which protein is incubated with vesicles containing [^3^H]cholesterol, we obtain a stoichiometry (mol sterol/mol protein) at equilibrium of 0.11 ± 0.02 for Lam4S2 (see legend of Figure 6D) and 0.47 ± 0.20 (mean ± SD (n=7)) for Osh4 (data not included). The difference in equilibrium binding stoichiometries for the two proteins under our assay conditions is likely a reflection of their structural differences. The reviewer will appreciate that it is difficult to obtain KD values from such an assay, but it is clear that the relative K_D_ of Lam4S2 and Osh4 for cholesterol is only a few-fold different.

Second, as noted by the reviewer, DHE loading into the sterol binding pocket of Osh and Lam proteins can be measured by FRET-mediated changes in the fluorescence of local tryptophan residues. The reviewer cited the Tong et al. paper in error: this paper reports that Osh3 does NOT bind sterol. However, Osh4 binds sterol and this has been demonstrated by many authors using different techniques (see results of our measurement above) included the DHE-based FRET assay. Previously, as well as in the revised manuscript, we used the DHE-based FRET assay to demonstrate sterol binding to Lam4S2 (Gatta et al., 2015; Figure 1—figure supplement 2B).

Third, we previously reported difficulties in fully loading DHE into Lam4S2 (Jentsch et al., 2018), despite the spectral change upon DHE binding that we (Gatta et al., 2015) observe in the FRET-based assay. The DHE assay has never been used to report a stoichiometry of binding as the spectra are difficult to deconvolute – the best use of the assay is to monitor changes in DHE environment as described in the paper of de Saint Jean et al., 2011. Because of the difficulty in quantitative sterol loading, we ultimately reported the crystal structure and NMR HSQC spectrum of protein loaded with 25HC (Jentsch et al., 2018).

Fourth, related to the previous point, the difficulty in quantitative loading makes the release assay suggested by the reviewer hard to interpret; indeed, it is not clear how such an assay can be used to support or evaluate the MD data as there would be no way to load the K89A mutant with sterol.

Fifth, we have taken the reviewer's suggestion to use rates from DHE transport assays to infer the energy barrier for sterol extraction. This information is presented in the third paragraph of the Discussion.

- Authors should check at least that single-mutation K89A or S181A does not affect the membrane-binding ability of Lam4S2

We now include data to show that introduction of the point mutations (K89A or S181A) does not affect the ability of Lam4S2 to bind membranes. Thus, under our standard assay conditions the percentage of Lam4S2 bound to membranes is – WT 42 ± 7.2 (WT), 43 ± 3.0 (S181A), 49 ± 4.5 (K89A) (mean ± standard deviation (n=8)). This information is reported in the legend of Figure 6D.

- it is quite surprising that 2 mutants out of 3 are not defective in yeast. In particular the D61A mutant do not phenocopy the K89A mutant, which is unexpected considering the existence of a electrostatic interaction with K89 and its central position in the triad of residues, making also contact with S181. Is there any explanation for this ?

To address the question, we carried out MD simulations of the D61A Lam4S2 mutant in which we specifically concentrated on how polar interactions with residue K89 are modified due to this mutation. As shown in Author response image 1, we find that in the wild type system, K89 mostly interacts with D61, while in the mutant protein the sidechain of K89 interacts instead with the carbonyl group of the adjacent N63. K89 can also interact with S181 as well as with water molecules, in both the WT and the D61A mutant. These results suggest that in the absence of the aspartic acid side chain at position 61, the dynamics of the binding site side-entrance are regulated by interactions between K89 and N63, resulting in wild type-like phenotype. While this argument is reasonable, it is speculative at this stage and more extensive computations will be needed to understand fully the functional mechanism of the D61A mutant. For this reason, we chose not to include this discussion and figure in the revised manuscript.

**Author response image 1. respfig1:** Snapshots of cholesterol-Lam4S2 complex from MD simulations of the wild type protein(left) and the D61A mutant (right), highlighting residue K89 and its interacting partners. In the wild type system, K89 interacts with D61, whereas in the mutant it interacts with N63.

Reviewer #3:[…] I suggest addressing the following question in more detail: Sterol molecules are amphipathic. The binding pocket for sterols must accommodate both polar and hydrophobic regions of cholesterol. How does the binding pocket deal with water influx into hydrophobic areas of the binding pocket? Is there evidence for structural changes in the protein to reduce energetically unfavorable interactions?

We find that the sterol binding cavity in Lam4S2 domain is surprisingly enriched in polar amino acids. Indeed, as we show in Figure 1—figure supplement 1B, the β strands that line the wall of the cavity contain multiple polar/charged residues (see green (polar), blue (positive), red (negative) colors). This includes the β4 strand that harbors Q121 which coordinates the sterol hydroxyl group in the X-ray structure of Lam4S2 and in our MD simulations. The sterol binding sites in other Lam/GramD1 domains have similarly strong polar characteristics. This finding supports our water-mediated mechanism of sterol release from these domains. This point is highlighted in the Introduction (last paragraph) as well as in Figure 1—figure supplement 1B as noted above.